# PARAPHRASE-ROBUST CONFORMAL PREDICTION FOR RELIABLE LLM UNCERTAINTY QUANTIFICATION

## ABSTRACT

Uncertainty quantification (UQ) provides interpretable measures of predictive confidence and supports reliable decision-making with large language models (LLMs). However, existing UQ methods are often neither statistically rigorous nor robust to paraphrase variations. To address these limitations, we propose a new framework for paraphrase-robust UQ, which builds on conformal prediction to ensure valid coverage and introduces a paraphrase-aware nonconformity score to enhance robustness. The score is derived by generating independent semantic paraphrases of each query, training an ancillary model that both approximates and robustifies the predictive distribution, and aggregating variability across these paraphrases. On five general multiple-choice Question Answering (MCQA) datasets and two medical MCQA datasets with Qwen2.5-7B, our method achieves nominal coverage with compact prediction sets and demonstrates improved robustness across different rewording settings. The results also generalize to Llama-3.1-8B and Phi-3-small, underscoring the reliability of the framework across model families. Code is available at https://anonymous.4open.science/r/paraphrase_uq-FDD8.

## 1 INTRODUCTION

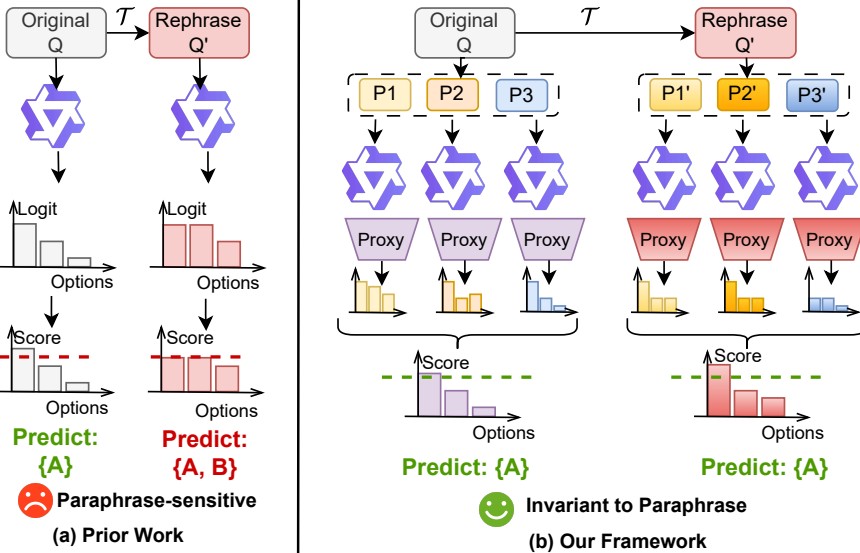

Figure 1: Prior scores is sensitive to rewording of input question (denoted as $\mathcal{T}$), producing larger prediction sets (left). Our PA framework aggregates multiple paraphrases with a proxy model to produce stable, paraphrase-invariant nonconformity scores (right).

Large language models (LLMs) have been rapidly deployed in high-stakes domains such as education and medicine (Bouchard & Chauhan, 2025; López et al., 2025). Despite their impressive performance, LLMs often exhibit overconfidence: the probabilities in their outputs do not reliably

reflect the true uncertainty of their predictions (Shorinwa et al., 2025). This miscalibration poses serious risks in safety-critical applications, where decision-makers need to know how uncertain the model prediction is. However, common token-level heuristics (e.g., entropy, margins, logit ranks) provide ad hoc uncertainty estimates without statistical guarantees (Shorinwa et al., 2025; Nado et al., 2022; Ulmer et al., 2022; Band et al., 2024; Huang et al., 2024b), which limits their reliability in practice.

To ensure statistical validity, conformal prediction (CP) provides a principled wrapper: given any nonconformity score, CP constructs prediction sets with finite-sample coverage under mild exchangeability assumptions (Vovk et al., 2005; Shafer & Vovk, 2008). Due to this task-agnostic guarantee, CP has been applied to many LLM tasks such as multiple-choice question answering (QA), factuality evaluation, and generation alignment (Quach et al., 2024a; Gui et al., 2024; Ye et al., 2024; Wang et al., 2024b; Su et al., 2024). A key challenge, however, lies in choosing a reliable nonconformity score. Ideally, a nonconformity score should be stable under paraphrasing: since natural language admits many equivalent expressions, such rewordings should not cause large fluctuations in the prediction set. For example, the questions "What is the capital of France?" and "Which city serves as France's capital?" should yield comparable uncertainty sets.

In practice, however, even semantically equivalent rewording of input question can induce distribution shifts in model's predictive behaviors, especially when the reworded questions are of low quality or introduce subtle syntactic differences (see the left panel in Figure 1). Although CP still guarantees valid coverage in theory when the calibration and evaluation samples remain exchangeable, different nonconformity scores behave very differently in finite samples, often producing unstable or inflated sets. For instance, as shown in Table 1, the prediction set size for two popular CP scores, LAC (Sadinle et al., 2019) and APS (Romano et al., 2020), nearly doubles under reworded inputs compared to clean inputs. This exposes a fundamental gap in existing CP score design, which has largely overlooked robustness to paraphrasing.

In this work, we ask whether it is possible to design a nonconformity score that remains robust under paraphrasing. We provide an affirmative answer by introducing *paraphrase-aware* nonconformity scores that explicitly enforce semantic invariance. As illustrated in the right panel of Figure 1, our pipeline first generates a diverse set of paraphrases for each input question, embeds them into a shared semantic space, and then trains a lightweight proxy model that produces calibrated predictive probabilities across paraphrases. The resulting scores aggregate variability across paraphrases and are inherently robust to rewordings of input questions. When applied to both split CP and the finer quasi-conditional CP (QCCP) (Gibbs et al., 2025), these scores consistently preserve target coverage while substantially reducing prediction set sizes.

We summarize our contributions as follows:

- **New evaluation setting.** We introduce prompt rewording as a new setting for evaluating LLM uncertainty, and show that previous conformal methods degrade in performance even under semantically equivalent rewordings.
- **Paraphrase-aware scores.** We propose *paraphrase-aware nonconformity scores* that can be applied within both split CP and QCCP, maintaining formal coverage guarantees while yielding smaller sets under paraphrasing.
- **Theoretical analysis.** We introduce distributional closeness to formally characterize the effect of paraphrasing, and we show that our scores preserve score-level exchangeability even under the distributional shift induced by paraphrased inputs.
- **Large-scale validation.** We conduct large-scale experiments on five general QA and two medical QA benchmarks, showing that our method consistently achieves nominal coverage with up to $2$–$4\times$ smaller sets than existing baselines, together with detailed ablations and analyses.

## 2 RELATED WORK

**Heuristic and Calibration-Based Uncertainty for LLMs.** LLM uncertainty is often estimated from token-level signals such as entropy, logits, or ranks, followed by calibration. Early work analyzes calibration of deep models and NLP tasks (Nado et al., 2022; Ulmer et al., 2022; Si et al., 2022), and extends to prompt- or generation-level schemes for long-form outputs (Band et al., 2024; Huang et al., 2024b). Recent methods replace raw probabilities with representation-based surrogates, in-

troducing semantic entropy probes (Kossen et al., 2024), relevance-aware confidence for free-form generation (Duan et al., 2024), perturbation-based measures (Gao et al., 2024), multi-agent diversity signals (Feng et al., 2024), and semantic-density metrics (Qiu & Miikkulainen, 2024). Complementary directions include abstention mechanisms (Madhusudhan et al., 2025), multicalibration for confidence scores (Detommaso et al., 2024), post-hoc calibration from generated text (Ulmer et al., 2024), and parameter-efficient Bayesian or ensemble-style methods for fine-tuned LLMs (Balabanov & Linander, 2025; Wang et al., 2024a). While these methods improve empirical calibration, they offer neither distribution-free coverage nor robustness to paraphrasing. In contrast, our approach introduces paraphrase-aware scores that seamlessly integrates conformal prediction to provide coverage guarantees.

**Conformal Prediction for LLMs.** Conformal prediction (CP) provides distribution-free prediction sets with finite-sample coverage under exchangeability (Vovk et al., 2005; Shafer & Vovk, 2008). Recent work has adapted CP to language modeling in several ways, including conformal language modeling (Quach et al., 2024a;b), API-only inference without logit access (Su et al., 2024), and schemes tailored to multiple-choice questions (Kumar et al., 2023; Yang & Liu, 2025). CP has also been applied to align and certify outputs (Gui et al., 2024), as well as to benchmark LLMs with uncertainty metrics beyond accuracy (Ye et al., 2024). Extensions to generation tasks include `ConU`, which applies split CP to sets of sampled responses (Wang et al., 2024b), and `SConU`, which analyzes exchangeability violations to approximate conditional guarantees (Wang et al., 2025b). Further refinements combine CP with re-asking strategies to improve accuracy and compactness (Vishwakarma et al., 2025), while selective answering with risk control has been explored through conformal abstention (Tayebati et al., 2025; Wang et al., 2025a). Our approach instead enforces *semantic invariance* via paraphrase-robust scores and leverages *quasi-conditional calibration* on semantic embeddings, yielding stronger coverage guarantees than the marginal coverage provided by standard split CP.

**Toward Conditional Guarantees and Paraphrase Robustness.** Exact conditional coverage is unattainable in finite samples without distributional assumptions (Vovk, 2012; Foygel Barber et al., 2021), which has motivated relaxations such as *quasi-conditional* guarantees via augmented quantile regression (Gibbs et al., 2025). For LLMs, recent extensions of CP exploit feature-conditional structure (Cherian et al., 2024). In parallel, another line of work emphasizes *semantic* rather than purely lexical uncertainty, introducing embedding-based metrics and perturbation procedures (Gao et al., 2024; Kossen et al., 2024; Huang et al., 2024a). Related approaches probe prompt sensitivity and meaning-preserving perturbations (Qiu & Miikkulainen, 2024; Cox et al., 2025), though they do not provide conformal guarantees. Our method integrates these two directions: (i) quantifying predictive stability across paraphrases using sentence embeddings and a proxy classifier, and (ii) calibrating these scores with QCCP (Gibbs et al., 2025), thereby producing prediction sets that are both semantically robust and statistically valid.

## 3 PARAPHRASE-ROBUST QUASI-CONDITIONAL CONFORMAL PREDICTION

**Problem Setup.** LLMs often produce predictions that are brittle to paraphrasing and hard to calibrate. Our goal is to construct *prediction sets* that not only guarantee statistical coverage but also remain stable under meaning-preserving rewordings. Formally, we consider supervised prediction with input $x \in \mathcal{X}$ (e.g., a natural language question) and a finite label space $\mathcal{Y}$ (e.g., multiple choice answers). Each example $(X_i, Y_i)$ provides a ground-truth label $Y_i \in \mathcal{Y}$. Our goal is to construct a prediction set $\widehat{C}(x) \subseteq \mathcal{Y}$ that maintains guaranteed coverage when the input $x \in \mathcal{X}$ is being paraphrased, under both senses of split CP and quasi-conditional CP (QCCP).

### 3.1 PARAPHRASE-AWARE NONCONFORMITY SCORES

**Method Overview.** Our method introduces a *paraphrase-aware* (PA) nonconformity score that captures semantic stability by aggregating predictions across paraphrases. The PA score can be integrated with either standard split CP or QCCP, which provides robustness to reworded inputs while ensuring statistically valid coverage through the conformal prediction component. At a high level (Figure 2), our approach first generates paraphrases for each question and extracts the corresponding LLM hidden states. A lightweight *proxy classifier* with calibration-aware training then maps these hidden states to confidence probability estimates, from which paraphrase-aware nonconfor-

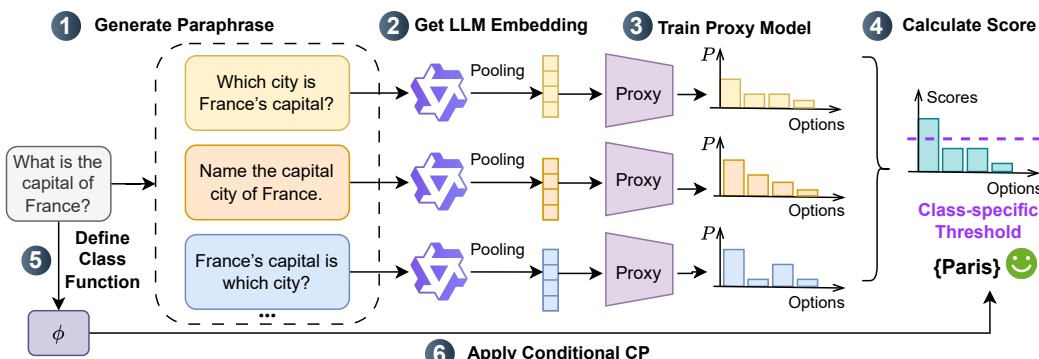

Figure 2: Schematic illustration of our proposed PA score integrated with QCCP.

mity scores are derived. Finally, the scores are calibrated with either split CP or QCCP to produce the prediction set for the given question. Due to space limit, a formal algorithm is provided in Algorithm 1 in the Appendix. We evaluate the paraphrase quality of the LLM paraphrase generator in Appendix F.7. Additionally, we provide runtime analysis for our pipeline in Appendix F.8.

**Learning a Proxy Classifier for Well-calibrated LLM Uncertainty.** Because raw LLM logits may be over-confident and ill-calibrated, we introduce a lightweight proxy classifier $P_\theta(y \mid E(x))$ that approximates the LLM's decision behavior. This proxy takes the last hidden state of the LLM as input and outputs a confidence probability distribution over the labels $\mathcal{Y}$. Implemented as a shallow two-layer MLP, it is trained on LLM embeddings $\{E(X_i)\}$ with labels $\{Y_i\}$ using a soft-binned ECE loss (Karandikar et al., 2021) as calibration loss in addition to task loss (cross-entropy for classification task). This design mitigates the poor calibration of raw LLM probabilities while remaining computationally efficient. When the proxy is well calibrated, its outputs are also easy to interpret. For example, if it assigns $P_\theta(\texttt{Paris} \mid E(x)) = 0.8$, this means that roughly 80% of question paraphrases with similar embeddings have "Paris" as the correct answer.

**Remark 3.1.** *We provide empirical evidence in Appendix F.9 showing that the PA score is not sensitive to a particular embedding choice. We further evaluate alternative pooling and layer configurations (including mean pooling, intermediate-layer representations, and attention-weighted pooling) and find that the PA score is stable across all variants.*

**Three Paraphrase-aware Nonconformity Scores.** We now turn to the construction of nonconformity scores, the core component of our method. A standard choice in CP is the probability-based score (Sadinle et al., 2019)

$$S_{\mathrm{prob}}(x,y) = 1 - P_\theta(y \mid E(x)),$$

which simply measures the lack of confidence in assigning label $y$ to question $x$ under the proxy model $P_\theta$. While this baseline captures predictive uncertainty for a single prompt, it neglects the instability that often arises when the same question is rephrased. To address this limitation, we define three paraphrase-aware nonconformity scores. From the previous paraphrasing step, we collect a paraphrase set $\mathcal{B}(x) = \{x'_1, \ldots, x'_m\}$ for each question $x$. Based on this set, the first score

$$S_{\mathrm{mean}}(x,y) = \frac{1}{m} \sum_{x' \in \mathcal{B}(x)} S_{\mathrm{prob}}(x',y),$$

averages the base scores across paraphrases, capturing overall semantic stability. The second,

$$S_{\mathrm{weighted}}(x,y) = \frac{\sum_{x' \in \mathcal{B}(x)} w(x,x') \, S_{\mathrm{prob}}(x',y)}{\sum_{x' \in \mathcal{B}(x)} w(x,x')}, \quad w(x,x') = \exp\big(-\mathrm{sim}(E(x), E(x'))\big),$$

assigns greater weight to paraphrases that are closer in the embedding space, thereby emphasizing local semantic similarity. The third takes the maximum score across paraphrases

$$S_{\mathrm{worst}}(x,y) = \max_{x' \in \mathcal{B}(x)} S_{\mathrm{prob}}(x',y)$$

which provides a measure that is sensitive to the hardest rephrasing.

Mean, weighted, and worst-case scores represent a spectrum from efficiency to robustness. Mean yields the most compact sets but may miss rare paraphrases. Weighted emphasizes closer variants to

balance compactness and robustness. The worst-case score is sensitive to low-quality paraphrases. We compare all three and quantify their trade-offs in Figure 7 and analyze their properties in detail in Appendix F.10.

## 3.2 SPLIT CONFORMAL AND QUASI-CONDITIONAL CONFORMAL CALIBRATION

To ensure statistically valid coverage, we apply either split CP or its refinement, quasi-conditional CP (QCCP) (Gibbs et al., 2025), on top of our PA nonconformity scores.

**Split CP.** Given an i.i.d. calibration set $\{(X_i, Y_i)\}_{i=1}^n$, split CP constructs the prediction set for a new input $X_{n+1}$ as

$$\widehat{C}(X_{n+1}) = \{y : S(X_{n+1}, y) \leq \widehat{\gamma}_\alpha\}, \tag{1}$$

where $S$ is a nonconformity score (cf. Section 3.1) and $\widehat{\gamma}_\alpha$ is the empirical $(1 - \alpha)$ quantile of $\{S(X_i, Y_i)\}_{i=1}^n \cup \{\infty\}$. Split CP is simple and distribution-free, but it relies on a single global threshold $\widehat{\gamma}_\alpha$ for all new inputs $X_{n+1}$, which can be overly conservative in heterogeneous regions of the input space.

**Quasi-conditional CP (QCCP).** QCCP refines split CP by adapting conformity thresholds to inputs. In doing so, it achieves guarantees that interpolate between marginal and conditional coverage. More specifically, given a precollected i.i.d. calibration set $\{(X_i, Y_i)\}_{i=1}^n$, it first computes scores $S_i = S(X_i, Y_i)$ for each $i = 1, \ldots, n$. These scores are then used to fit the following augmented quantile regressor,

$$\widehat{g}_S \in \arg\min_{g \in \mathcal{F}} \frac{1}{n+1} \Big[ \sum_{i=1}^n \ell_\alpha\big(g(X_i), S_i\big) + \ell_\alpha\big(g(X_{n+1}), S\big) \Big], \tag{2}$$

where $\mathcal{F} = \{\phi(\cdot)^\top \beta : \beta \in \mathbb{R}^d\}$ denotes the linear class over the class mapping $\phi$ (which we will specify later) and $\ell_\alpha$ is the pinball loss, defined by $\ell_\alpha(\theta, S) = (S - \theta)(\mathbf{1}_{\{S \geq \theta\}} - \alpha)$. At inference time, for a new pair $(X_{n+1}, Y_{n+1})$ produced by the LLM, we construct the prediction set

$$\widehat{C}(X_{n+1}) = \big\{y : S(X_{n+1}, y) \leq \widehat{g}_{S(X_{n+1},y)}(X_{n+1})\big\}. \tag{3}$$

The advantage of QCCP lies in the introduction of $\widehat{g}_S$, which adaptively estimates thresholds based on the class function $\phi$, rather than relying on the fixed global threshold $\widehat{\gamma}_\alpha$ used in split CP. Consequently, QCCP provides stronger coverage guarantees than the marginal coverage by split CP (see Appendix A for details).

**Choice of the Class Function $\phi$.** We now specify the choice of the class function $\phi$. At a high level, we want to group questions by their semantic type (e.g., commonsense vs. factual), so that QCCP can calibrate thresholds within semantically coherent classes and avoid the conservativeness of a single global threshold.

However, in practice, most benchmarks lack explicit type annotations, so we construct them automatically. Specifically, we embed each question–context pair using a pretrained SBERT encoder (`all-MiniLM-L6-v2`) (Reimers & Gurevych, 2019) to obtain sentence embeddings, and then perform $K$-means clustering on the embeddings of the calibration and test splits. Each sample is assigned to the nearest centroid, and the resulting cluster assignments serve as $\phi(x)$. Mathematically, this amounts to partitioning $\mathcal{X}$ into groups $\{\mathcal{G}_j\}_{j=1}^m$ (with $m$ the total number of clusters) and setting $\phi(x) = \sum_{j=1}^m \mathbf{1}\{x \in \mathcal{G}_j\} \cdot \phi_j$, where $\phi_j$ denotes the representative vector of cluster $j$. The hyperparameter $K$ can either be fixed or automatically selected via the silhouette score on calibration embeddings. In short, our choice of $\phi$ uses sentence embeddings to generate semantically coherent clusters and allows QCCP to condition on latent question classes without requiring manual labels.[1]

## 3.3 THEORETICAL ANALYSIS

Classic CP relies on exchangeability, and its coverage guarantees fail under distribution shifts. In settings where we can actively impose the shift and the used operation is distributionally closed, we can restore exchangeability by applying the operation to all data. We formalize this intuition below.

---

[1]For more implementation details, see Appendix D.

**Assumption 3.1** (Distributional closedness)**.** *Let $\mathcal{T} : \mathcal{X} \rightarrow \mathcal{X}$ be a random map that is independent of the input. Moreover, $\mathcal{T}$ is distributionally closed, meaning that for any input $x$, $\mathcal{T} \circ \mathcal{T}(x) \stackrel{d}{=} \mathcal{T}(x)$.*

**Proposition 3.1** (Data exchangeability under distribution shift)**.** *Suppose Asm. 3.1 holds, and let $\{(X_i, Y_i)\}_{i=1}^{n+1}$ be an exchangeable (or i.i.d.) dataset. $(\mathcal{T}(X_{n+1}), Y_{n+1})$ might not be exchangeable with $(X_i, Y_i)_{i=1}^{n}$ due to the distribution shift induced by $\mathcal{T}$. However, applying $\mathcal{T}$ to all inputs restores exchangeability, i.e., $\left(\mathcal{T} \circ \mathcal{T}(X_{n+1}), Y_{n+1}\right)$ is exchangeable (or i.i.d.) with $\left\{(\mathcal{T}(X_i), Y_i)\right\}_{i=1}^{n}$.*

**Lemma 3.1** (Score exchangeability)**.** *Let $\{(X_i, Y_i)\}_{i=1}^{n+1}$ be an i.i.d. dataset. Then our porposed scores $S$, i.e., $S \in \{S_{\text{mean}}, S_{\text{weighted}}, S_{\text{worst}}\}$, is exchangeable in the following settings:*

- *Normal setting: $S(X_{n+1}, Y_{n+1})$ is exchangeable with $\{S(X_i, Y_i)\}_{i=1}^{n}$.*
- *Fully reworded setting: $S(\mathcal{T}(X_{n+1}), Y_{n+1})$ is exchangeable with $\{S(\mathcal{T}(X_i), Y_i)\}_{i=1}^{n}$.*
- *Semi-reworded setting: Under Asm. 3.1, $S(\mathcal{T}(X_{n+1}), Y_{n+1})$ is exchangeable with $\{S(X_i, Y_i)\}_{i=1}^{n}$.*

This requirement of distribitional closedness is natural for paraphrasing, since paraphrasing a sentence and then paraphrasing it again typically preserves its semantic content, with only surface-level variations arising from the redundancy of natural language. As a result, our proposed scores are able to handle the distributional shift introduced by paraphrasing. For each input $x$, our score uses $m$ independent samples from $\mathcal{T}(x)$ to construct a robust statistic, and the exchangeability result above continues to hold for the vector formed by these $m$ samples. Consequently, we can recover marginal or quasi-conditional coverage when applying our score within CP or QCCP, even under the rewording distributional shift. A formal statement is provided in Appendix A.

## 4 EMPIRICAL PERFORMANCE

**Datasets.** We evaluate our framework on seven benchmark datasets. Five of them are general multiple-choice question answering (MCQA) datasets from the LLM-Uncertainty-Benchmark (Ye et al., 2024), including MMLU, CosmosQA, HellaSwag, HaluDial, and HaluSum. They correspond to different tasks: question answering (**QA**), reading comprehension (**RC**), commonsense inference (**CI**), dialogue response selection (**DRS**), and document summarization (**DS**). Each dataset contains 10,000 MCQA questions. In addition, we include two medical MCQA datasets, **MedMCQA** (Pal et al., 2022) and **MedQA** (Jin et al., 2021), and sample 10,000 questions from each. Although we focus on MCQA datasets for evaluation, we discuss extension to other tasks in Appendix F.11.

**Baselines and Evaluation Metrics.** We compare our method with two strong nonconformity score baselines: Least Ambiguous set-valued Classifiers (LAC) (Sadinle et al., 2019) and Adaptive Prediction Sets (APS) (Romano et al., 2020). We justify the choice of baselines in Appendix F.6. Data are split into training, calibration, and test sets with a 40/30/30 ratio. Evaluation on the test set uses two metrics: **Coverage Rate (CR)**, the fraction of examples where the true label is included in the prediction set, and **Set Size (SS)**, the average number of labels in the prediction set. Unless otherwise specified, the target coverage is $1 - \alpha = 0.90$.

**Reworded Input Question Generation.** To stress-test our PA score, we generate reworded inputs by prompting a local LLM to rephrase each question while preserving its semantics. We introduce distributional shifts through stochastic generation (temperature sampling) and post-process outputs to ensure they remain valid, well-formed questions. When the model fails to produce a valid rewording, we retry with a stricter template or fall back to a simple rule-based rewrite. Additional implementation details are provided in Appendix B. We further show in Appendix F.4 that our findings remain stable when using a different paraphrase generator, `Llama-3.1-8B-Instruct`, instead of `Qwen2.5-7B-Instruct`. The quality of the generated rewordings is evaluated in Appendix F.7.

**Practical setup for the three evaluation settings.** We next describe how the three settings in Section 3.3 are instantiated in our experiments. ▷ **Normal setting.** All training, calibration, and test examples use the original human-written questions. No paraphrasing is applied on the input question level. ▷ **Fully-reworded setting.** We apply the same paraphrasing operator $\mathcal{T}$ (reworded by an LLM) to *every* input in the training, calibration, and test sets. All CP scores are computed on these LLM-reworded questions. ▷ **Semi-reworded setting.** Training and calibration sets remain

human-written input questions, but at test time the input questions are reworded by an LLM. More technical details for the Semi-reworded setting are in Appendix F.5.

## 4.1 PA IS ROBUST TO REWORDINGS ACROSS DATASETS AND CP METHODS

We first evaluate whether our proposed PA score remains robust under prompt rewording, where each test question is rephrased using the algorithm described above. On `Qwen2.5-7B-Instruct` (Qwen et al., 2025), we apply different scores to both split CP and QCCP across five MCQA benchmarks (Table 1). We observe that the PA score maintains coverage tightly around the $90\%$ target on both normal and fully reworded settings, while producing compact prediction sets. By contrast, APS consistently overshoots coverage (95–97%), indicating unreliable guarantees, and LAC, although close to nominal coverage, exhibits large set-size inflation (from $3.44$ to $4.79$ under the fully reworded setting).

Table 1: Strong performance of the proposed PA nonconformity scores under split CP and QCCP with `Qwen2.5-7B-Instruct` . Results are reported on normal and fully reworded (Full) test sets across five benchmarks. Bold numbers indicate the best.

| Method | QA Normal | QA Full | RC Normal | RC Full | CI Normal | CI Full | DRS Normal | DRS Full | DS Normal | DS Full |
|---|---|---|---|---|---|---|---|---|---|---|
| *Coverage Rate (CR, %) Better if closer to $90\%$ — Split CP* | | | | | | | | | | |
| LAC | **89.63** | 88.43 | 89.00 | **90.33** | 91.80 | 91.27 | **89.80** | **90.37** | 89.23 | 90.43 |
| APS | 97.97 | 99.03 | 93.00 | 92.27 | 95.60 | 91.70 | 99.07 | 90.90 | 92.30 | 91.87 |
| PA | **89.63** | **89.23** | **89.77** | 90.67 | **91.10** | **90.57** | 90.77 | 90.77 | 91.77 | 90.93 |
| *Coverage Rate (CR, %) Better if closer to $90\%$ — QCCP* | | | | | | | | | | |
| LAC | **90.07** | 88.47 | 88.87 | 90.63 | 91.63 | 91.00 | 95.73 | **90.13** | **89.70** | **90.13** |
| APS | 96.10 | **90.20** | 93.00 | 92.50 | 98.87 | 92.17 | 92.67 | 91.40 | 97.93 | 91.67 |
| PA | 89.67 | 89.47 | **90.10** | **90.33** | **91.23** | **90.33** | **90.87** | 91.00 | 91.90 | 91.13 |
| *Set Size (SS) ↓ — Split CP* | | | | | | | | | | |
| LAC | 3.13 | 4.79 | 3.41 | 3.87 | 1.94 | 4.18 | 3.43 | 4.51 | 2.42 | 3.47 |
| APS | 5.45 | 5.92 | 4.20 | 4.19 | 3.44 | 4.27 | 5.78 | 4.51 | 3.34 | 3.82 |
| PA | **2.25** | **2.71** | **1.19** | **1.32** | **1.53** | **1.97** | **1.56** | **2.12** | **1.00** | **1.47** |
| *Set Size (SS) ↓ — QCCP* | | | | | | | | | | |
| LAC | 3.12 | 4.76 | 3.40 | 3.91 | 2.04 | 4.10 | 4.96 | 4.49 | 2.51 | 3.44 |
| APS | 5.05 | 5.19 | 4.20 | 4.22 | 5.40 | 4.32 | 4.22 | 4.59 | 5.20 | 3.82 |
| PA | **2.25** | **2.71** | **1.21** | **1.30** | **1.65** | **1.97** | **1.58** | **2.13** | **1.00** | **1.49** |

To further assess robustness under distribution shifts induced by paraphrasing, we also evaluate all methods in the semi-reworded (Semi.) setting and show the results in Table 2. Across all five benchmarks, PA attains coverage closest to $90\%$ while maintaining the smallest prediction sets, indicating that PA effectively mitigates distribution shift even when only the test-time inputs are reworded. These results show that our PA score offers two key advantages: robustness to input-question distribution shifts and compact prediction sets.

Table 2: Strong performance of the proposed PA nonconformity scores under the **semi-reworded (Semi.)** setting. Bold numbers indicate the best.

| | Method Semi. | QA | RC | CI | DRS | DS | QA | RC | CI | DRS | DS |
|---|---|---|---|---|---|---|---|---|---|---|---|
| | | Coverage Rate (CR, %) Better if closer to 90% | | | | | Set Size (SS) ↓ | | | | |
| *Split CP* | LAC | 96.13 | 83.70 | 92.63 | 99.67 | **95.60** | 5.51 | 3.12 | 4.46 | 5.93 | 4.36 |
| | APS | 96.30 | **91.03** | 95.50 | 99.50 | 97.67 | 5.69 | 4.02 | 4.93 | 5.89 | 4.69 |
| | PA | **87.10** | 87.03 | **89.60** | **90.07** | 83.60 | **2.73** | **1.21** | **2.22** | **2.27** | **2.06** |
| *QCCP* | LAC | 94.50 | 83.77 | 87.70 | 99.90 | 96.20 | 5.35 | 3.11 | 3.56 | 5.98 | 4.45 |
| | APS | 93.77 | **91.00** | 98.80 | 95.53 | 99.27 | 5.52 | 4.02 | 5.74 | 5.19 | 5.60 |
| | PA | **86.33** | 87.40 | **88.50** | **89.97** | 83.60 | **2.71** | **1.23** | **2.17** | **2.28** | **2.06** |

## 4.2 GENERALIZATION ACROSS LLMS

Next, we evaluate cross-model generalization using two models, `Llama-3.1-8B-Instruct` (Dubey et al., 2024) and `Phi-3-small-8k-Instruct` (Abdin et al., 2024), on MMLU under

both normal and fully reworded settings. The results mirror the patterns observed in Section 4.1: APS tends to inflate coverage, often overshooting the nominal 90% level, while LAC achieves target coverage but suffers from inflated set sizes under rewording. In contrast, our score (PA) consistently produces the most compact prediction sets and aligns well with the nominal coverage.

Table 3: Cross-LLM generalization on MMLU. Coverage Rate (CR, %) closer to $90\%$ is better; Set Size (SS) ↓ is better. Results are shown for normal vs. fully reworded inputs (Full).

| | | Llama-3.1-8B | | | | Phi-3-small | | | |
| | | CR (%) | | SS | | CR (%) | | SS | |
| | **Method** | Normal | Full | Normal | Full | Normal | Full | Normal | Full |
| | LAC | 90.97 | **89.97** | 3.65 | 5.09 | 98.53 | 98.67 | 5.54 | 5.74 |
| *Split CP* | APS | 93.87 | 100.0 | 3.95 | 6.00 | 98.13 | 97.97 | 5.51 | 5.67 |
| | PA | **89.23** | 88.53 | **2.51** | **2.81** | **89.27** | **89.70** | **1.97** | **2.59** |
| | LAC | 91.10 | **90.30** | 3.67 | 5.12 | 95.87 | 88.47 | 4.64 | 4.28 |
| *QCCP* | APS | 93.87 | 99.97 | 3.96 | 6.00 | 91.03 | 75.83 | 3.96 | 2.76 |
| | PA | **89.03** | 88.97 | **2.51** | **2.85** | **89.23** | **90.00** | **1.96** | **2.61** |

## 4.3 ROBUSTNESS ON MEDICAL MCQA DATASETS

Finally, we evaluate whether paraphrase-aware conformal prediction generalizes to domain-specific settings by testing on two medical MCQA benchmarks, MedMCQA (Pal et al., 2022) and MedQA (Jin et al., 2021), using Qwen2.5-7B-Instruct under both normal and fully reworded conditions. As shown in Table 4, the results mirror the patterns observed in the general domain: APS variants often overshoot or fluctuate around the 90% target (e.g., $99.6\%$ on MedQA), while LAC stays closer to nominal coverage but produces inflated sets when reworded inputs are introduced. In contrast, our PA score consistently maintains coverage near the target and yields the most compact prediction sets (e.g., 2.6–3.1 on MedMCQA vs. 4.4–4.6 for LAC/APS). These findings confirm that PA robustly preserves both coverage and efficiency even in the medical domain, demonstrating its ability to generalize beyond general-purpose benchmarks.

Table 4: Results on two medical QA benchmarks with Qwen2.5-7B-Instruct. Coverage rate (CR, %) closer to the nominal 90% is better, while smaller set size (SS, ↓) denotes better.

| | | MedMCQA | | | | MedQA | | | |
| | | CR (%) | | SS | | CR (%) | | SS | |
| | **Method** | Normal | Full | Normal | Full | Normal | Full | Normal | Full |
| | LAC | 89.43 | **89.50** | 4.46 | 4.48 | 91.67 | **89.43** | 4.47 | 5.00 |
| *Split CP* | APS | 91.30 | 91.67 | 4.47 | 4.45 | 99.60 | 88.80 | 5.85 | 4.72 |
| | PA | **89.67** | 91.47 | **2.59** | **3.01** | **89.73** | 89.13 | **4.02** | **4.31** |
| | LAC | **90.03** | **90.43** | 4.55 | 4.60 | 91.23 | 89.87 | 4.64 | 5.02 |
| *QCCP* | APS | 92.03 | 91.50 | 4.59 | 4.52 | 93.67 | 89.57 | 4.92 | 4.81 |
| | PA | 90.43 | 91.00 | **2.77** | **3.09** | **89.90** | **89.90** | **4.03** | **4.39** |

## 5 IN-DEPTH ANALYSIS AND ABLATION STUDIES

**Proxy Model Improves Accuracy and Calibration.** In our work, the proxy model $P_\theta$ is a lightweight two-layer MLP trained on top of frozen hidden states from the LLM. We argue that $P_\theta$ yields more accurate and better-calibrated distributions than raw LLM logits. As shown in Figure 3, this proxy consistently outperforms logits across benchmarks on all calibration metrics. For example, accuracy improves substantially (**CI** $0.263 \to 0.728$, **QA** $0.287 \to 0.551$), while both Brier score and NLL decrease (**RC** Brier $0.126 \to 0.050$, NLL $1.592 \to 0.647$). Here, Brier captures the mean squared error between predicted probabilities and true one-hot labels, while NLL penalizes models that assign low probability to the correct answer, both measuring the calibration quality. This

improvement arises because raw logits are optimized for next-token prediction rather than calibrated posteriors, which leads to miscalibration and poor class separation. In contrast, the proxy leverages hidden states, which encode richer task-relevant signals, and is trained with a calibration-aware loss. As a result, it has better accuracy and calibration.

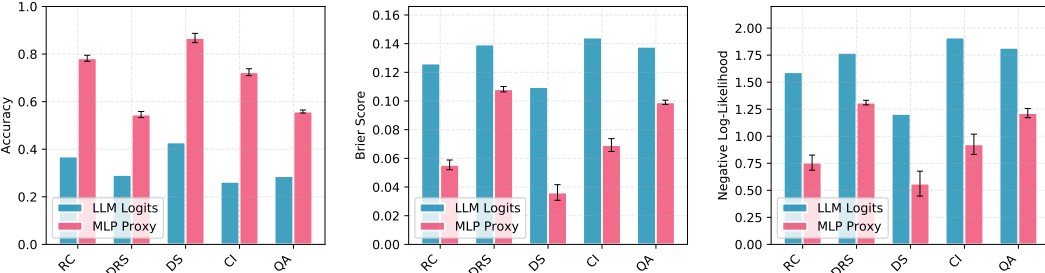

Figure 3: The proxy improves both task performance and calibration consistently. Evaluation of prediction accuracy and calibration metric using proxy model vs. raw LLM logits across datasets. Left: accuracy (higher is better). Middle: Brier (lower is better). Right: NLL (lower is better).

**Proxy Model Decreases Prediction Set Size.** Next, we examine the effect of the proxy model on prediction set size. To this end, we ablate the proxy model by computing the PA score directly from LLM logits. Across all five datasets, coverage remains close to the nominal 90%, but *set sizes increase substantially* without the proxy under both split CP (in Appendix, Figure 8) and QCCP (in Figure 4). This shows that the proxy produces better-calibrated predictive distributions, which translate into materially smaller sets at comparable coverage.

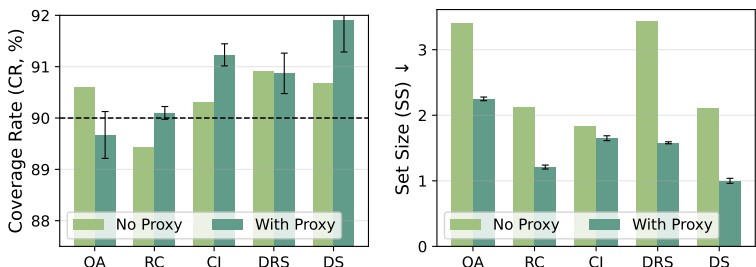

Figure 4: Effect of removing the proxy model under the quasi-conditional CP setting. Coverage remains close to 90% (dashed line), but set size increases notably without the proxy model.

**Better Class-conditional Coverage.** We group questions into semantic classes (or clusters) and evaluate class-conditional coverage by visualizing empirical coverage rates on different classes. As shown in Figure 5, PA consistently outperforms LAC and APS across different classes.

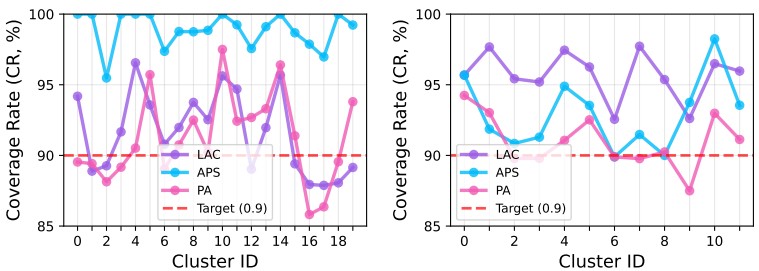

Figure 5: Evaluation of class-conditional coverage with QCCP on the HellaSwag dataset (left) and HaluDial dataset (right). Results on additional datasets are provided in Figure 9.

**Effect of Coverage Levels.** To examine the effect of the level $\alpha$, we vary $\alpha \in \{0.2, 0.05, 0.01\}$ and evaluate both coverage and prediction set size again. As shown in Figure 6, our PA score consistently tracks the nominal $(1 - \alpha)$ target while maintaining compact sets. For example, at $\alpha = 0.05$, it achieves $\approx 95\%$ coverage with an average set size of 2.7, compared to APS's inflated

5.0. In contrast, APS again overshoots, while LAC produces smaller but unstable sets. Overall, PA consistently outperforms across different levels.

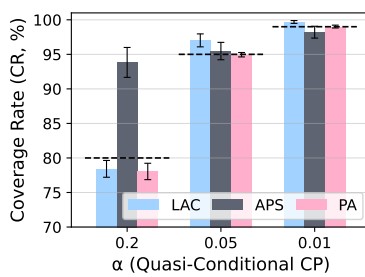 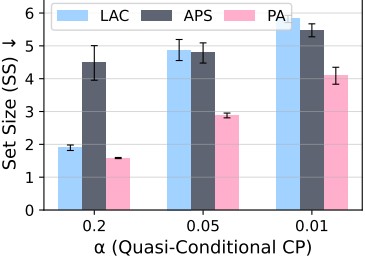

Figure 6: Coverage rate (left) and prediction set size (right) vs. $\alpha$ on MMLU under QCCP.

**Larger Paraphrase Budgets Yield Smaller Prediction Sets.** We now study the effect of the paraphrase budget $m$ for `Qwen2.5-7B-Instruct` (Qwen et al., 2025) on the MMLU dataset. As shown in Table 5, increasing $m \in \{2, 4, 6\}$ leads our PA score to achieve higher coverage and smaller set sizes. This is intuitive: more paraphrases provide richer semantic views of each question, which reduces variance in the aggregated score and enables more precise calibration.

**Comparison of Different Scores.** Finally, we compare the three PA scores, namely Mean, Weighted, and Worst, introduced in Section 3.1. We evaluate them under both split CP and QCCP settings. As shown in Figure 7, the Mean score consistently yields the most compact sets (whose SS $\approx 1.5$–$1.6$) while staying close to the target $90\%$ coverage. The Weighted variant produces moderately larger sets (whose SS $\approx 2.3$) without noticeable gains in coverage, whereas the Worst variant greatly inflates set size (whose SS $\approx 3.0$) and overshoots coverage (which

Table 5: Effect of paraphrase budget $m$.

|  | $m{=}2$ | $m{=}4$ | $m{=}6$ |
| --- | --- | --- | --- |
| *Coverage Rate (CR, %)* | | | |
| PA (marginal) | 89.10 | 89.47 | **89.63** |
| PA (conditional) | **89.87** | 89.60 | 89.67 |
| *Set Size (SS)* | | | |
| PA (marginal) | 2.42 | 2.32 | **2.25** |
| PA (conditional) | 2.49 | 2.34 | **2.25** |

$\geq 95\%$). QCCP mitigates some of the overshoot for Mean and Weighted but largely preserves their relative ranking. Overall, Mean offers the best efficiency, Weighted provides only a mild trade-off, and Worst proves overly conservative.

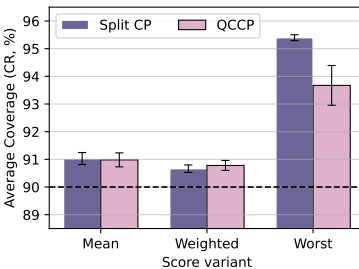 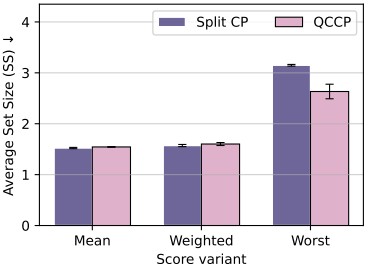

Figure 7: Comparison of different scores. Left: average coverage (closer to 90% is better). Right: average set size (smaller is better).

# 6 CONCLUSION

In this work, we introduced a framework for paraphrase-robust conformal prediction by designing paraphrase-aware nonconformity scores and applying them to both split CP and quasi-conditional CP. Our method preserves theoretical coverage guarantees while yielding substantially smaller prediction sets than logit-based baselines. Experiments on five general QA and two medical QA benchmarks demonstrate that it remains reliable under rewording and generalizes across model families. More broadly, our work illustrates how CP can be adapted to address semantic invariance and distribution shifts in LLM uncertainty quantification. Promising directions include extending paraphrase-robust scores to free-form generation, integrating them with selective abstention policies, and exploring theoretical bounds under broader perturbation classes.

ETHICAL STATEMENT

All authors have read and adhere to the conference Code of Ethics. We acknowledge the use of large language models (LLMs) for limited purposes in this paper, only for polishing the writing and assisting with literature search. All LLM-generated content was carefully reviewed and verified by the authors, who take full responsibility for the final manuscript.

REPRODUCIBILITY STATEMENT

To support reproducibility, we provide an anonymized GitHub repository link at the end of the abstract containing our codebase. Detailed descriptions of the dataset are included in Appendix C, and all hyperparameter settings are reported in Appendix E.

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

---

**Algorithm 1:** Paraphrase-Robust Quasi-Conditional CP

---

**Input:** Calibration set $\{(X_i, Y_i)\}_{i=1}^n$, paraphrase generator, embedding map $E$, proxy model
       $P_\theta$, class function $\phi$, confidence level $\alpha$, score $S$ (mean, weighted, or worst-case).

**Train proxy:** Fit $P_\theta$ on $\{(E(X_i), Y_i)\}$ using a calibration-aware loss.

**Compute scores:** For each $(X_i, Y_i)$, compute $S_i \leftarrow S(X_i, Y_i)$ and record class $\phi(X_i)$.

**for** each test question $x$ **do**
> Initialize $\widehat{C}(x) \leftarrow \emptyset$.
> Generate a set of paraphrases $\mathcal{B}(x)$ and compute embeddings $\{E(x')\}_{x' \in \mathcal{B}(x)}$.
> Determine class label $z \leftarrow \phi(x)$.
> **for** $y \in \mathcal{Y}$ **do**
> > Compute paraphrase-aware score $S(x, y)$.
> > Query the class-conditional quantile function $\widehat{g}_z$ from QCCP.
> > Add $y$ to $\widehat{C}(x)$ if $S(x, y) \leq \widehat{g}_z(x)$.
>
> **return** $\widehat{C}(x)$.

---

## A    COVERAGE GUARANTEES

### A.1    PRELIMINARIES

**Theoretical guarantee of Split CP.**

**Lemma A.1** ((Vovk et al., 2005))**.** *Assume $(X_i, Y_i)_{i=1}^{n+1}$ are i.i.d. and that $S(X, Y)$ has a continuous distribution. The prediction set from split CP is*

$$\widehat{C}(X_{n+1}) := \{y : S(X_{n+1}, y) \leq Quantile_\alpha(\{S(X_i, Y_i)\}_{i=1}^n)\}$$

*where $Quantile_\alpha(\{S(X_i, Y_i)\}_{i=1}^n)$ is the empirical $(1-\alpha)$ quantile of $\{S(X_i, Y_i)\}_{i=1}^n \cup \{\infty\}$. Then, it achieves marginal coverage:*

$$1 - \alpha \ \leq \ \mathbb{P}(Y_{n+1} \in \widehat{C}(X_{n+1})) \ \leq \ 1 - \alpha + \frac{1}{n+1}. \tag{4}$$

**Theoretical guarantee of Quasi-Conditional CP.**    Lemma A.2 establishes the quasi-conditional coverage guarantee for QCCP. This guarantee is strictly stronger than the marginal coverage of split conformal prediction (the case where $\mathcal{F}$ contains only constant functions) and strictly weaker than full conditional coverage (which would require $\mathcal{F}$ to be all measurable functions), providing a principled middle ground. In practice, computing equation 3 reduces to a convex optimization; see Section 4 of Gibbs et al. (2025) for details.

**Lemma A.2** (Theorem 2 in (Gibbs et al., 2025))**.** *Let $\mathcal{F} = \{\phi(\cdot)^\top \beta : \beta \in \mathbb{R}^d\}$ denote the class of linear functions over the basis $\phi : \mathcal{X} \rightarrow \mathbb{R}^d$. Assume $(X_i, Y_i)_{i=1}^{n+1}$ are i.i.d. and that $S(X, Y) \mid X$ has a continuous distribution. Let the prediction set be*

$$\widehat{C}(X_{n+1}) := \{y : S(X_{n+1}, y) \leq \widehat{g}_{S(X_{n+1}, y)}(X_{n+1})\}$$

*where $\widehat{g}_S$ is solved from equation 2. Then, for any $f \in \mathcal{F}$,*

$$\left| \mathbb{E}\left[ f(X_{n+1}) \left( \mathbf{1}\{Y_{n+1} \in \widehat{C}(X_{n+1})\} - (1 - \alpha) \right) \right] \right| = \frac{d}{n+1} \cdot \mathbb{E}\left[ \max_{1 \leq i \leq (n+1)} |f(X_i)| \right].$$

### A.2    FORMAL THEORETICAL RESULTS AND PROOFS

**Proposition A.1** (Restatement of Proposition 3.1)**.** *Suppose Asm. 3.1 holds, and let $\{(X_i, Y_i)\}_{i=1}^{n+1}$ be an exchangeable (or i.i.d.) dataset. $(\mathcal{T}(X_{n+1}), Y_{n+1})$ might not be exchangeable (or i.i.d.) with $(X_i, Y_i)_{i=1}^n$ due to the semantic shift induced by $\mathcal{T}$. However, applying $\mathcal{T}$ to all inputs restores exchangeability (or i.i.d.) , i.e., $(\mathcal{T} \circ \mathcal{T}(X_{n+1}), Y_{n+1})$ is exchangeable (or i.i.d.) with $\{(\mathcal{T}(X_i), Y_i)\}_{i=1}^n$.*

*Proof of Proposition A.1.* If $Z_i = (X_i, Y_i)$, exchangeability means $(Z_{\pi(1)}, \ldots, Z_{\pi(n+1)}) \overset{d}{=} (Z_1, \ldots, Z_{n+1})$ for any permutation $\pi$. Applying the same map to each coordinate preserves exchangeability, so the sequence $\{(\mathcal{T}(X_i), Y_i)\}_{i=1}^{n+1}$ is exchangeable. Hence $(\mathcal{T}(X_{n+1}), Y_{n+1})$ is exchangeable with $\{(\mathcal{T}(X_i), Y_i)\}_{i=1}^{n}$. By Assumption 3.1, $\mathcal{T}(\mathcal{T}(x)) \overset{d}{=} \mathcal{T}(x)$ for all $x$, and therefore $\mathcal{T}(\mathcal{T}(X_{n+1})) \overset{d}{=} \mathcal{T}(X_{n+1})$. Thus $(\mathcal{T} \circ \mathcal{T}(X_{n+1}), Y_{n+1}) \overset{d}{=} (\mathcal{T}(X_{n+1}), Y_{n+1})$. Replacing the last coordinate of an exchangeable vector by an identically distributed variable does not change its joint law. Therefore $(\mathcal{T} \circ \mathcal{T}(X_{n+1}), Y_{n+1})$ is exchangeable with $\{(\mathcal{T}(X_i), Y_i)\}_{i=1}^{n}$. The proof for the i.i.d. case is the same and thus omitted. $\square$

**Notations.** Fix any data point $(X, Y)$ from either the calibration set or the test set. Our PA score first generates $m$ paraphrased versions of $X$ using a language model. Let $\mathcal{T}(X)$ denote the distribution of a single paraphrase, and let

$$Z^{(1)}, \ldots, Z^{(m)} \overset{\text{i.i.d.}}{\sim} \mathcal{T}(X).$$

After paraphrasing, the pair $(X, Y)$ is augmented into $(Z^{(1)}, \ldots, Z^{(m)}, Y)$. Since the calibrated predictor $P_\theta$ is trained on an unrelated dataset, it is treated as fixed throughout. Define the augmented feature

$$\widetilde{X} := (Z^{(1)}, \ldots, Z^{(m)}), \qquad \widetilde{\mathcal{T}}(X) := \widetilde{X}.$$

Therefore, the PA score can be written as

$$S_{\text{PA}}(X, Y) = S(Z^{(1)}, \ldots, Z^{(m)}, Y) = S(\widetilde{\mathcal{T}}(X), Y) = S(\widetilde{X}, Y) \tag{5}$$

for some deterministic computation rule $S$. This notation highlights that the PA score simply operates on the augmented data $(\widetilde{X}, Y)$.

**Lemma A.3** (Exchangeability for augmented data). *Let $\{(X_i, Y_i)\}_{i=1}^{n+1}$ be an i.i.d. dataset and $\{(\widetilde{\mathcal{T}}(X_i), Y_i)\}_{i=1}^{n+1}$ be the augmented data defined above. It follows that*

- ***Normal setting:*** $(\widetilde{\mathcal{T}}(X_{n+1}), Y_{n+1})$ *is i.i.d. with* $\{(\widetilde{\mathcal{T}}(X_i), Y_i)\}_{i=1}^{n}$.

- ***Fully reworded setting:*** $(\widetilde{\mathcal{T}}(\mathcal{T}(X_{n+1})), Y_{n+1})$ *is i.i.d. with* $\{(\widetilde{\mathcal{T}}(\mathcal{T}(X_i)), Y_i)\}_{i=1}^{n}$.

- ***Semi-reworded setting:*** *Under Asm. 3.1,* $(\widetilde{\mathcal{T}}(\mathcal{T}(X_{n+1})), Y_{n+1})$ *is i.i.d. with* $\{(\widetilde{\mathcal{T}}(X_i), Y_i)\}_{i=1}^{n}$.

With Lemma A.3, we can prove Lemma 3.1.

*Proof of Lemma 3.1.* By Lemma A.3, each proposed score $S \in \{S_{\text{mean}}, S_{\text{weighted}}, S_{\text{worst}}\}$ operates on the corresponding augmented data $(\widetilde{X}, Y)$. Since the augmented dataset remains i.i.d., the resulting scores are also i.i.d. across data points. Therefore, the scores are exchangeable. $\square$

Now, we provide the proof for Lemma A.3 below.

*Proof of Lemma A.3.* By construction, each entry of the random vector $\widetilde{\mathcal{T}}(X)$ is generated by i.i.d. sampling from $\mathcal{T}(X)$, and is independent across data points. Therefore, the transformed dataset $\{(\widetilde{\mathcal{T}}(X_i), Y_i)\}_{i=1}^{n+1}$ is still i.i.d., which proves the first point.

For the second point, by a similar argument, $\{\widetilde{\mathcal{T}}(\mathcal{T}(X_i)), Y_i)\}_{i=1}^{n+1}$ is i.i.d. because $\{(\mathcal{T}(X_i), Y_i)\}_{i=1}^{n+1}$ is i.i.d.

For the third point, Proposition A.1 implies that $(\mathcal{T} \circ \mathcal{T}(X_{n+1}), Y_{n+1})$ is i.i.d. with $\{(\mathcal{T}(X_i), Y_i)\}_{i=1}^{n}$. Each entry of $(\widetilde{\mathcal{T}}(\mathcal{T}(X_{n+1})), Y_{n+1})$ consists of i.i.d. samples from the distribution of $(\mathcal{T}(\mathcal{T}(X)), Y)$, which equals the distribution of $(\mathcal{T}(X), Y)$ by Assumption 3.1. On the other hand, by definition, each entry of $(\widetilde{\mathcal{T}}(X_{n+1}), Y_{n+1})$ is an i.i.d. sample from the same distribution. Therefore, $(\widetilde{\mathcal{T}}(\mathcal{T}(X_{n+1})), Y_{n+1})$ is i.i.d. with $\{(\widetilde{\mathcal{T}}(X_i), Y_i)\}_{i=1}^{n}$.

$\square$

Now, we are ready to state and prove the formal coverage results.

**Theorem A.1** (Formal coverage guarantee)**.** *Assume* $(X_i, Y_i)_{i=1}^{n+1}$ *are i.i.d. and that both* $S(X, Y)|X$ *and* $S(X, Y)$ *has a continuous distribution. Let* $\mathcal{T}$ *be an operator satisfying Assumption 3.1. We consider the following three data settings:*

- *Normal setting: The calibration dataset is* $\{(X_i, Y_i)\}_{i=1}^{n}$ *and the test data is* $(X_{n+1}, Y_{n+1})$.

- *Fully reworded setting: The calibration dataset is* $\{(\mathcal{T}(X_i), Y_i)\}_{i=1}^{n}$ *and the test data is* $(\mathcal{T}(X)_{n+1}, Y_{n+1})$.

- *Semi-reworded setting: The calibration dataset is* $\{(X_i, Y_i)\}_{i=1}^{n}$ *and the test data is* $(\mathcal{T}(X_{n+1}), Y_{n+1})$.

*For the above three data settings, the following coverage guarantees hold.*

- *PA score for CP. Let* $\widehat{C}_{\mathrm{CP}}(\cdot)$ *denote the prediction set produced by CP and defined in equation 1. It follows that*

$$1 - \alpha \ \leq \ \mathbb{P}(Y_{n+1} \in \widehat{C}_{\mathrm{CP}}(X_{n+1})) \ \leq \ 1 - \alpha + \frac{1}{n+1}.$$

- *PA score for QCCP. Let* $\mathcal{F} = \{\phi(\cdot)^{\top}\beta : \beta \in \mathbb{R}^d\}$ *denote the class of linear functions over the feature vector* $\phi : \mathcal{X} \to \mathbb{R}^d$. *Let* $\widehat{C}_{\mathrm{QCCP}}(\cdot)$ *be the prediction set produced by QCCP and defined in equation 2. Then, for any* $f \in \mathcal{F}$,

$$\left| \mathbb{E}\left[ f(X_{n+1}) \left( \mathbf{1}\{Y_{n+1} \in \widehat{C}_{\mathrm{QCCP}}(X_{n+1})\} - (1 - \alpha) \right) \right] \right| \leq \frac{d}{n+1} \cdot \mathbb{E}\left[ \max_{1 \leq i \leq (n+1)} |f(X_i)| \right].$$

*Proof of Theorem A.1.* The theorem follows directly by plugging Lemma A.3 into either Lemma A.1 or Lemma A.2. □

## B  PARAPHRASE GENERATION FOR FULL/SEMI-PARAPHRASE SETTINGS

We only use this procedure to generate paraphrases for the input question used in the full- and semi-paraphrase evaluation settings. This set of reworded questions is not directly used for calculating the PA score, but is used as another set of input questions.

**Prompt Pool.** We maintain 7 short templates (e.g., "Rephrase this question using varied vocabulary and phrasing: {question}"). One template is sampled per input to induce lexical or syntactic variety without changing semantics.

**Batched Generation.** Given a batch of $n$ questions, we form $n$ prompts and generate once with: `temperature=0.7`, `top_p=0.9`, `do_sample=True`, `eos/pad_token_id` aligned. We decode with `skip_special_tokens=True`.

**Cleaning & Validation.** We remove the prompt prefix and extract the first question sentence via regex matching the earliest "?". We then strip boilerplate ("Rephrase:", "Paraphrase:", "Here's a . . . "), quotes/bullets/code blocks, and normalize whitespace. The candidate paraphrase must satisfy: (i) non-empty, (ii) case-insensitive $\neq q$, (iii) ends with "?". Duplicates within the batch are dropped.

**Retry & Fallback.** Failures are retried with a stricter template: single line, no preface, $\leq 20$ words, must end with "?". Remaining failures are rewritten by a deterministic rule set that preserves meaning (e.g., "Which of the following"→"Which option", add trailing "?", etc.).

---

**Algorithm 2:** Paraphrase Pipeline (per question $q$)

---
1: Sample a paraphrase template; compose prompt $p(q)$.
2: Generate with ($T{=}0.7$, top-p=0.9, sampling).
3: Decode; strip prompt prefix; take first sentence ending with "?".
4: Clean and validate; if valid, return $\tilde{q}$.
5: Else: retry with strict one-line template; clean and validate.
6: Else: apply rule-based fallback; return $\tilde{q}$.

---

## C    DATASET DETAILS

We used five general and two medical MCQA datasets to show that our paraphrase-aware score can be broadly used and can adapt to high-stakes scenarios where calibration is important. The five general datasets test a wide range of LLM capabilities and have been used in previous CP benchmarking papers (Ye et al., 2024; Vishwakarma et al., 2025). We use the five general datasets processed by Ye et al. (2024). Each of the datasets contains 10,000 questions.

**QA, MMLU** (Hendrycks et al., 2021): MMLU is a dataset designed to test the general knowledge and *question-answering* abilities of LLMs. Question topics range from sociology and high school geography to electrical engineering and abstract algebra.

**RC, CosmosQA** (Huang et al., 2019): CosmosQA focuses on gauging an LLM's *reading comprehension* abilities. The LLM is given a short paragraph and is then asked to answer a follow-up question based on commonsense reasoning.

**CI, HellaSwag** (Zellers et al., 2019): HellaSwag evaluates if LLMs can use *commonsense inference* to construct a realistic and meaningful continuation of a given scenario. HellaSwag is deliberately designed so that LLMs struggle with questions that humans could normally answer with high confidence.

**DRS, HaluEval** (Li et al., 2023): HaluEval consists of hallucinated LLM responses to user queries. A subset of HaluEval contains queries that relate to *dialogue response selection*: the LLM must be able to choose a logical response for a conversation. We refer to this part of HaluEval as HaluDial.

**DS, HaluEval** (Li et al., 2023): The HaluEval dataset also has hallucinated *document summaries*. In an MCQA setting, the LLM must determine which summary in the answer choices is most relevant to a provided document. We refer to this part of HaluEval as HaluSum.

**MedMCQA** (Pal et al., 2022): MedMCQA is a large-scale multiple-choice medical QA dataset comprising over 194,000 entrance-exam–style questions spanning 2400 healthcare topics and 21 medical subjects. We select a subset of 10,000 single-answer questions from MedMCQA (i.e. where exactly one option is marked correct) for our experiments.

**MedQA** (Jin et al., 2021): MedQA is a medical exam QA dataset derived from professional medical board exams (e.g. USMLE), providing each question paired with candidate answer options and corresponding references. In our work, we use only the US-part of MedQA, and further restrict to 10,000 multiple-choice items that have exactly one correct answer.

## D    DETAILED METHOD IMPLEMENTATION

**Paraphrase generation for calculating PA score.** Paraphrases were generated by prompting `Qwen2.5-7B` (Qwen et al., 2025) with the following query: "Rephrase the following question in your own words (preserving its meaning): Original question: {question} \n Rephrased question:". Additional details, such as any context or the answer choices, were not included with the original question. A total of 6 paraphrases were generated per question. Paraphrases that were equivalent to the original question or were previously generated were not included in the final set. The temperature of `Qwen2.5-7B` (Qwen et al., 2025) was set to 1.1 to encourage diverse responses.

**LLM embeddings.** We next obtained the LLM embeddings for each question and paraphrase. These LLM embeddings are used to train the proxy model and serve as the LLM's representation of the query. We treated the paraphrases as new samples in the dataset and assigned them the same answer choices and correct label as their parent question. For each sample, the input prompt included the question and the list of answer choices without the correct label. The LLM embedding was extracted from the final hidden layer; this layer simultaneously encodes the LLM's understanding of the question and its predicted answer. The LLM logits were also retrieved by finding the raw score corresponding to each answer represented as a token (e.g. 'A', 'B', etc.). We then applied softmax to the logits to obtain a probability distribution over the answer choices. For each dataset except the DS dataset, the LLM was provided with 2 example questions and their correct labels. Due to the long context needed for the questions in the DS dataset, only 1 example was provided to the LLM in this case. We followed this procedure to get the embeddings and logits of the instruction-

tuned versions of `Qwen2.5-7B` (Qwen et al., 2025) , `Llama-3.1-8B` (Dubey et al., 2024), and `Phi-3-small-8k` (Abdin et al., 2024).

**Proxy model.** The proxy model calibrates the LLM's probability distribution of the answer choices across the paraphrases. We used a 2-layer MLP with ReLU activation as our proxy model. The input to the proxy model is an LLM's embedding for a question or paraphrase, and the output is a vector with dimension $|\mathcal{Y}|$. The loss function used to train the proxy model is $L_{total} = L_{CE} + \lambda_{ece} \times L_{ECE}$, where $L_{CE}$ is the standard cross-entropy loss for multiclass classification and $L_{ECE}$ is the soft-binned expected calibration error loss (Karandikar et al., 2021). $\lambda_{ECE}$ is a hyperparameter that can be tuned to increase or decrease the relative importance of calibration in different contexts. Out of the 4,000 questions in the training set, 600 questions were used as the validation set. Since each question has 6 paraphrases, the effective training and validation set sizes are 23,800 and 4,200, respectively. We conducted grid search on the following hyperparameters: learning rate, weight decay, $\lambda_{ECE}$, hidden dimension, and the batch size. The hyperparameters used for each dataset and each model are provided in Appendix E.

**Score calculation.** For our baselines, we only use logits from the *original* question. The LAC score (Sadinle et al., 2019) is defined as $s_{\text{LAC}}(x, y) = 1 - f(x)_y$, where $f(x)_y$ is the softmax probability of label $y$ for a question $x$. The APS score (Romano et al., 2020) is $s_{\text{APS}}(x, y) = \sum_{y' \in \mathcal{Y}: \, f(x)_{y'} \geq f(x)_y} f(x)_{y'}$, i.e., the cumulative probability of labels ranked at least as high as $y$. To find the score of a question $x$ using the PA method, the LLM embeddings of its paraphrases (represented by the set $\mathcal{B}(x)$) are inputted into the trained proxy model. Then, the score for answer choice $y$ of question $x$ is given by $S_{\text{mean}}(x, y)$, $S_{\text{weighted}}(x, y)$, or $S_{\text{worst}}(x, y)$ (see Section 3.1). Note that only the scores of the 6 paraphrases are used in the PA formulas. The LLM's embedding of the original question does not contribute to that question's final PA score. This is in contrast to LAC and APS, which only rely on the logits of the original question and do not factor in the paraphrases.

**Split conformal prediction.** Split CP (Vovk et al., 2005) uses a separate calibration dataset to calculate a global score threshold that determines the prediction sets for the test set. For each calibration example, we use the nonconformity score from the LLM's softmax probabilities or proxy model's probability distribution. These nonconformity scores are collected and the $(1 - \alpha)$ quantile is estimated with a finite-sample correction $q_\alpha = \text{Quantile}(\{s_i\}, \lceil (n+1)(1-\alpha) \rceil / n)$. At test time, each example's prediction set is formed by including all labels whose score is below $q_\alpha$, with a fallback to the most probable label if the set is empty. This procedure ensures that the empirical coverage approaches the nominal $1 - \alpha$ guarantee.

**Quasi-conditional conformal prediction.** In contrast to split CP, QCCP (Gibbs et al., 2025) calculates class-specific thresholds for pre-defined classes. A function $\phi$ is defined that assigns a class to a question based on its features. In the case that $\phi$ is a constant function, then QCCP is equivalent to split CP; we use this $\phi$ function ($\phi$ = intercept) to obtain all split CP results. In our QCCP analysis, we focus on a $\phi$ function that takes in the embedding of each question generated by `all-MiniLM-L6-V2` (Reimers & Gurevych, 2019). $K$-means clustering is used to separate the questions into clusters, with each cluster acting as a class, and a different score threshold is defined for each cluster. For a given test question, either a one-hot encoding of its assigned cluster or a vector of its embedding's distances to the three closest cluster centroids (dist3) can be used to calculate the threshold. The coverage rates reported using QCCP are calculated marginally with the exception of Figure 5 and Figure 9, which displays class-wise coverages. We used the `conditionalconformal` package (Gibbs et al., 2025) to implement QCCP.

# E  HYPERPARAMETER SETTINGS FOR REPRODUCIBILITY

Tables 6 and 7 list the framework hyperparameters for the five general MCQA datasets and the two medical MCQA datasets, respectively. Tables 8 and 9 report the corresponding hyperparameters for the proxy model on the general and medical datasets.

Table 6: Hyperparameters for QA, RC, CI, DRS, DS. We train an MLP proxy with ECE regularization and compute paraphrase-aware scores using 6 paraphrases per question. Evaluation includes QCP with SBERT-clustered $\Phi$ and plain CP with intercept $\Phi$, using the same manifest-based splits and $\alpha$=0.1.

| Setting / Hyperparameter | QA, RC, CI, DRS, DS |
|---|---|
| **General** | |
| Base LLM (for reps/logits) | Qwen2.5-7B-Instruct |
| Random seed | 42 |
| Samples per question ($n$) | 7 (1 base + 6 paraphrases) |
| Options per item ($|\mathcal{Y}|$) | 6 (A–F) |
| **Proxy model (training)** | |
| Input dim | 3584 |
| Architecture | MLP: $3584 \rightarrow h$ (ReLU) $\rightarrow 6$ |
| Optimizer | Adam |
| Max epochs / patience | 200 / 20 |
| Loss | CE + soft-binned ECE (15 bins, temp 0.1)) |
| **Paraphrase-aware score computation** | |
| Paraphrases per question | 6 |
| Metric used | `S_mean` |
| **Conformal prediction / QCP evaluation** | |
| Prompting / ICL | `base` / `icl1` |
| Error level ($\alpha$) | 0.1 |
| Split config | manifest: `tr0.4` / `cf0.3` / `tf0.3` |
| QCP $\Phi$ mode | `cluster_sbert` |
| SBERT model | MiniLM-L6-v2 |
| Cluster selection | auto |
| $\Phi$ representation | dist3 |
| Embedding norm / mini-batch k-means | on / on |
| Also reported (plain CP) | `intercept` $\Phi$ (same splits, $\alpha$) |

# F ADDITIONAL RESULTS FOR ANALYSIS AND ABLATION STUDIES

## F.1 PROXY MODEL ABLATION UNDER SPLIT CP

We report proxy model ablation results under split CP in Figure 8; the same trend holds under QCCP (Figure 4), where removing the proxy leaves coverage near 90% but substantially enlarges set sizes.

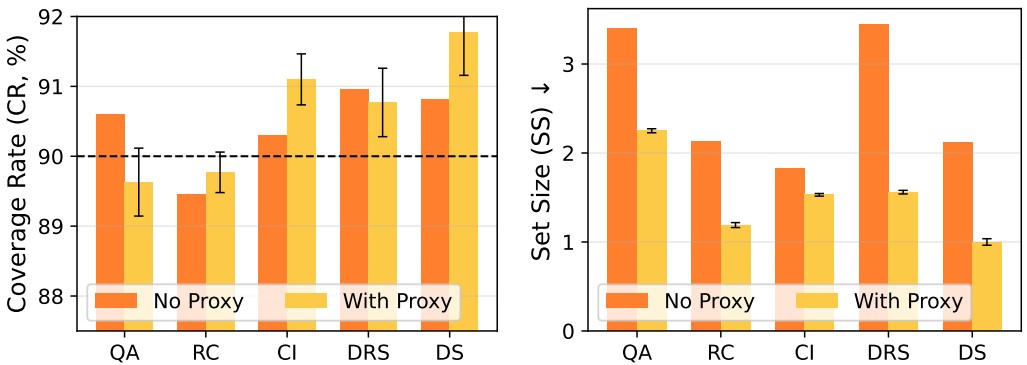

Figure 8: Effect of removing the proxy model under the split CP setting. Coverage remains close to 90% (dashed line), but set size increases notably without the proxy model.

Table 7: Hyperparameters for MedMCQA-10k and MedQA-10k. We train an MLP proxy with ECE regularization and compute paraphrase-aware scores using 6 paraphrases per question. Evaluation includes QCP with SBERT-clustered $\Phi$ and plain CP with intercept $\Phi$, using the same manifest-based splits and $\alpha=0.1$.

| Setting / Hyperparameter | MedMCQA-10k | MedQA-10k |
|---|---|---|
| **General** | | |
| Base LLM (for reps/logits) | Qwen2.5-7B-Instruct | |
| Random seed | 42 | |
| Samples per question ($n$) | 7 (1 base + 6 paraphrases) | |
| Options per item ($K$) | 6 (A–F) | |
| **Proxy model (training)** | | |
| Input dim | 3584 | |
| Architecture | MLP: $3584 \to h$ (ReLU) $\to 6$ | |
| Optimizer | Adam | |
| Max epochs / patience | 50 / 5 | |
| Loss | CE + soft-binned ECE (15 bins, temp 0.1) | |
| **Paraphrase-aware score computation** | | |
| Paraphrases per question | 6 | |
| Metric used | S_mean | |
| **Conformal prediction / QCP evaluation** | | |
| Prompting / ICL | task/icl1 | |
| Error level ($\alpha$) | 0.1 | |
| Split config | manifest: tr0.4 / cf0.3 / tf0.3 | |
| QCP $\Phi$ mode | cluster_sbert | |
| SBERT model | MiniLM-L6-v2 | |
| Cluster selection / $K$ | fixed / 20 | |
| $\Phi$ representation | one-hot (cluster ID) | |
| Embedding norm / mini-batch k-means | off / off | |
| Also reported (plain CP) | intercept $\Phi$ (same splits, $\alpha$) | |

Table 8: Proxy model specific hyperparameters for the 5 general datasets

| Dataset | Hidden dimension $h$ | Batch size | Learning rate | Weight decay | $\lambda_{\textbf{ECE}}$ |
|---|---|---|---|---|---|
| MMLU | 256 | 64 | 1e-3 | 0.0 | 0.5 |
| CosmosQA | 512 | 128 | 1e-3 | 0.0001 | 0.5 |
| HellaSwag | 256 | 128 | 1e-3 | 0.0001 | 0.5 |
| HaluDial | 256 | 64 | 1e-3 | 0.0 | 0.5 |
| HaluSum | 256 | 128 | 1e-4 | 0.0 | 0.5 |

Table 9: Proxy model specific hyperparameters for MedMCQA and MedQA.

| Dataset | Hidden dimension $h$ | Batch size | Learning rate | Weight decay | $\lambda_{\textbf{ECE}}$ |
|---|---|---|---|---|---|
| MedMCQA | 256 | 64 | 1e-3 | 0.0 | 0.5 |
| MedQA | 256 | 64 | 1e-3 | 0.0 | 0.5 |

### F.2 CLASS LEVEL EVALUATION FOR ADDITIONAL DATASETS

We report class-conditional coverage for the remaining datasets in Figure 9; the results mirror those in the main text, with PA consistently achieving better class-level coverage than LAC and APS across classes.

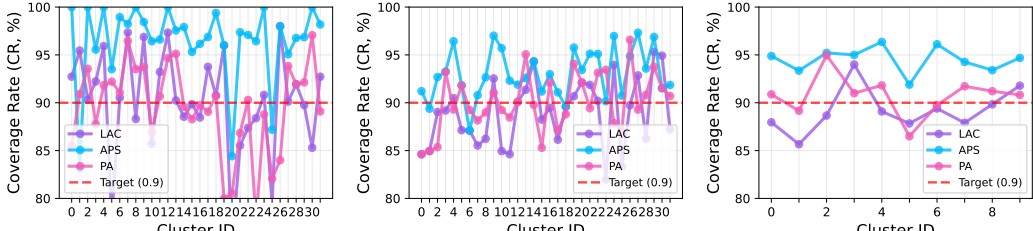

Figure 9: Evaluation of class-conditional coverage with QCCP on the MMLU (top left), CosmosQA, (top right), and HaluSum (bottom center) datasets.

## F.3 DIFFERENT RISK LEVEL ANALYSIS UNDER SPLIT CP

We report results at different user-specified risk levels under split CP in Figure 10. The trends mirror those of QCCP in the main text: PA tracks the nominal $(1 - \alpha)$ target more closely while keeping sets compact, whereas APS overshoots and LAC has larger set sizes for low values of $\alpha$.

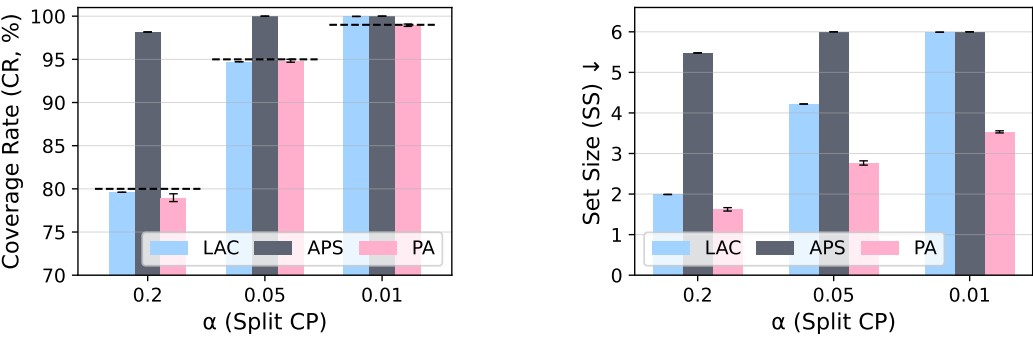

Figure 10: Coverage rate (left) and prediction set size (right) vs. $\alpha$ on MMLU under Split CP.

## F.4 A DIFFERENT LLM AS A PARAPHRASE GENERATOR

To assess the sensitivity of our framework to the choice of the paraphrase and rewording model, we use `Llama-3.1-8B` to generate paraphrases while keeping the rest of the pipeline fixed. In particular, we still use `Qwen2.5-7B` to extract the hidden embeddings. We evaluate this setup on three datasets (CosmosQA, HellaSwag, and MedMCQA) under both the normal and fully reworded conditions.

As shown in Table 10, our PA score demonstrates strong robustness to the change in paraphrase and rewording generator. Under split CP, the PA score maintains coverage rates close to the 90% target and yields smaller set sizes across all datasets. Similar trends are observed under QCCP.

## F.5 SEMI-REWORDED SETTING

Recall that the two versions of each dataset in our evaluation are defined as $\mathcal{D}_{\text{orig}} = \{(X_i, Y_i)\}_{i=1}^N$ and $\mathcal{D}_{\text{reworded}} = \{(\mathcal{T}(X_i), Y_i)\}_{i=1}^N$, where $\mathcal{T}$ represents the rewording operation. In the semi-reworded setting, we test a realistic semantic shift scenario where the calibration set is drawn from $\mathcal{D}_{\text{orig}}$ (human-written inputs), but the test set is drawn from $\mathcal{D}_{\text{reworded}}$ (LLM-reworded inputs). Under this shift, the exchangeability assumption required for standard conformal prediction is violated for baselines like LAC and APS, as the distribution of their nonconformity scores (which depend on raw logits) changes between calibration and testing.

However, we hypothesize that our PA score remains robust to this shift. Our PA score is calculated by aggregating predictions over the set of paraphrases. We denote this paraphrasing generation as $\mathcal{T}^*$ to distinguish between the rewording used to create a distinct dataset ($\mathcal{T}$) and the paraphrasing used to calculate the PA score ($\mathcal{T}^*$). Consequently, for a calibration point $X$, the score $S(X, Y)$ depends on $\mathcal{T}^*(X)$. For a test point $\mathcal{T}(X)$, the score depends on the composition $\mathcal{T}^*(\mathcal{T}(X))$. In-

Table 10: Performance of nonconformity scores under split CP and QCCP with `Llama-3.1-8B` as the paraphrase generator. Results are reported on normal and fully reworded test sets across three benchmarks. Bold numbers indicate the best performance.

| | RC (Cosmos) | | CI (HellaSwag) | | MedMCQA | |
|---|---|---|---|---|---|---|
| **Method** | Normal | Full | Normal | Full | Normal | Full |
| *Coverage Rate (CR, %) Better if closer to* $90\%$ *— Split CP* | | | | | | |
| LAC | 88.63 | 89.06 | **90.90** | **90.40** | **89.83** | **90.40** |
| APS | 99.30 | 93.30 | 93.53 | 98.87 | 99.87 | 99.83 |
| PA | **90.33** | **89.70** | 91.70 | 91.90 | 90.90 | 90.96 |
| *Coverage Rate (CR, %) Better if closer to* $90\%$ *— QCCP* | | | | | | |
| LAC | 89.03 | 89.03 | **91.06** | **91.27** | 96.83 | 91.17 |
| APS | 94.90 | 97.23 | 97.30 | 95.17 | 96.13 | 97.00 |
| PA | **90.23** | **89.80** | 92.13 | 91.90 | **90.70** | **91.03** |
| *Set Size (SS)* $\downarrow$ *— Split CP* | | | | | | |
| LAC | 2.81 | 3.69 | 3.37 | 4.23 | 3.57 | 5.03 |
| APS | 5.77 | 4.50 | 4.66 | 5.88 | 5.98 | 5.98 |
| PA | **1.16** | **1.30** | **1.70** | **1.96** | **2.55** | **2.98** |
| *Set Size (SS)* $\downarrow$ *— QCCP* | | | | | | |
| LAC | 2.84 | 3.76 | 3.39 | 4.42 | 5.19 | 5.10 |
| APS | 4.44 | 5.34 | 5.34 | 5.29 | 5.10 | 5.66 |
| PA | **1.15** | **1.49** | **1.72** | **2.22** | **2.56** | **3.10** |

voking Assumption 3.1 (distributional closedness), we posit that $\mathcal{T}^*(\mathcal{T}(X)) \overset{d}{\approx} \mathcal{T}^*(X)$. Therefore, the distribution of PA scores should remain invariant between the original and reworded inputs, preserving coverage guarantees even when the test data contain semantic shifts not observed during test time.

We further evaluate our hypothesis in the semi-reworded setting using `Llama-3.1-8B` as both the dataset-level rewriter ($\mathcal{T}$) and the internal paraphrase generator ($\mathcal{T}^*$). As shown in Table 11, LAC performs poorly under this shift, achieving only 87.17% coverage on CosmosQA and 86.97% on HellaSwag, both well below the nominal 90% target. APS overshoots on coverage and does so with substantially inflated prediction sets (SS up to 5.85 on CosmosQA and 4.79 on HellaSwag), making its predictions considerably less actionable.

In contrast, PA delivers the best tradeoff between reliability and efficiency. Under split CP, PA achieves 89.97% coverage on CosmosQA and 90.27% on HellaSwag—both tightly aligned with the nominal 90%—while producing the smallest prediction sets (SS $\approx$ 1.41–2.03). Under QCCP, PA again reaches coverage closest to 90% and maintains compact sets, substantially outperforming both LAC and APS. Compared with Table 2, these results highlight that the quality of the paraphrase generator $\mathcal{T}^*$ is crucial: with a stronger model such as `Llama-3.1-8B`, the semantic closure assumption is better satisfied, leading to more stable proxy training and more reliable PA uncertainty estimates.

### F.6 CLARIFICATION OF BASELINE COMPARISON

Our method is developed within the *conformal prediction (CP)* framework, where uncertainty is represented through *per-option nonconformity scores* used to construct *set-valued predictions* with finite-sample coverage guarantees. Therefore, only methods that produce compatible per-option scores and operate within CP provide a meaningful basis for comparison.

Table 11: Performance of nonconformity scores under the **semi-reworded (Semi.)** setting on `Llama-3.1-8B` as paraphrase generator. Coverage Rate (CR, %) closer to 90% is better; Set Size (SS) ↓ is better. Results are reported on RC (Cosmos) and CI (HellaSwag). Bold numbers indicate best performance.

| | Dataset | RC | CI | RC | CI |
|---|---|---|---|---|---|
| | Semi. | Coverage Rate (CR) | | Set Size (SS) | |
| *Split CP* | LAC | 87.17 | 86.97 | 3.51 | 3.91 |
| | APS | 99.33 | 90.90 | 5.85 | 4.79 |
| | PA | **89.97** | **90.27** | **1.41** | **2.03** |
| *QCCP* | LAC | 87.00 | 87.77 | 3.51 | 3.98 |
| | APS | 97.40 | 95.10 | 5.41 | 5.33 |
| | PA | **90.00** | **90.43** | **1.43** | **2.01** |

Heuristic LLM uncertainty measures (as discussed in Section 2) such as *semantic entropy* (Kossen et al., 2024) or *perturbation-based UQ* (Gao et al., 2024) output a *single scalar instance-level uncertainty score*, typically evaluated with AUC. These methods do not produce prediction sets, do not ensure coverage guarantees, and cannot be directly integrated into CP without redesigning the score structure. As their outputs and evaluation metrics are fundamentally different from CP (prediction sets with coverage/set-size tradeoffs), numerical comparison would not be meaningful.

We therefore compare against the two standard CP baselines, **LAC** (Sadinle et al., 2019) and **APS** (Romano et al., 2020), which (i) provide the same distribution-free coverage guarantees, (ii) use per-option scores compatible with CP, and (iii) are widely adopted in prior CP-for-classification work. Our results demonstrate that paraphrase-aware (PA) scores maintain valid coverage while substantially reducing prediction set size, particularly under paraphrasing shifts.

Finally, while recent CP work explores *learned* nonconformity scores (Einbinder et al., 2022; Xie et al., 2024; Kiyani et al., 2024), none addresses semantic robustness or paraphrase-induced shifts. Prior work on paraphrasing for UQ (Bakman et al., 2025) focuses on heuristic metrics without coverage guarantees. Our method is the first to introduce a learned, paraphrase-aware score design that preserves CP validity under meaning-preserving transformations while improving efficiency.

### F.7 PARAPHRASE QUALITY

To verify that the generated paraphrases approximately preserve the semantics of the original questions, we conduct a quantitative paraphrase quality evaluation across CosmosQA (RC), HellaSwag (CI), and MedMCQA (medical MCQA). For each dataset, we use the same paraphrase generation pipelines as in the main experiments and evaluate all paraphrases that are actually used for conformal prediction.

**Paraphrase sets and generators.** Recall that for each original question $x$ we use three paraphrase-related sets: (i) $\mathcal{T}^*(x)$, the set of paraphrases used for PA score computation in the normal setting; (ii) $\mathcal{T}(x)$, the fully reworded questions used in the fully reworded setting; and (iii) $\mathcal{T}^*(\mathcal{T}(x))$, the paraphrases of fully reworded questions used for PA score computation in the fully reworded setting. We instantiate these sets using two independent LLM paraphrase generators: `Llama-3.1-8B` and `Qwen2.5-7B`. All quality metrics are reported separately for each generator and each scenario.

**Metrics.** Given an original question $x$ and a paraphrase $p$, we compute three standard similarity metrics:

1. **SBERT cosine similarity.** We use the `all-MiniLM-L6-v2` SentenceTransformer to embed both $x$ and $p$, normalize the embeddings, and compute

$$\cos(x, p) = \langle \text{SBERT}(x), \text{SBERT}(p) \rangle.$$

This serves as a semantic similarity measure that is robust to minor lexical changes.

2. **BLEU.** We compute sentence-level BLEU scores using `sacrebleu` with effective order enabled, treating the original question as the reference and the paraphrase as the hypothesis.

3. **ROUGE-L.** We compute ROUGE-L F-measure using the `rouge_score` library with stemming enabled, again using the original question as the reference and the paraphrase as the candidate.

**Aggregation and reporting.** For each question $x$ and each paraphrase set (e.g., $\mathcal{T}^*(x)$), we first compute the metric values for all paraphrases $p \in \mathcal{T}^*(x)$ and then average them to obtain a single per-question score. We repeat this for SBERT cosine, BLEU, and ROUGE-L. At the dataset level, we report the mean and standard deviation across all questions that have at least one paraphrase in the corresponding set. This yields, for each dataset, scenario, and model, a tuple of the form

$$\text{metric} = \mu \pm \sigma,$$

which we summarize in Table 12.

**Findings.** Across CosmosQA and MedMCQA, we observe moderate to high SBERT cosine similarity (typically in the range 0.70–0.82), indicating that the paraphrases remain close in semantic space to the original questions while allowing stylistic variation. HellaSwag exhibits lower cosine scores (0.37–0.53), which is expected due to its longer, multi-sentence contexts and richer surface-level variability. BLEU and ROUGE-L follow consistent trends across datasets and paraphrase scenarios. Overall, these results suggest that the paraphrase sets $\mathcal{T}^*(x)$, $\mathcal{T}(x)$, and $\mathcal{T}^*(\mathcal{T}(x))$ largely preserve the semantics of the original questions and are suitable for use in our paraphrase-aware conformal prediction framework.

## F.8 RUNTIME ANALYSIS

To assess the practical overhead of our framework, we measure the average wall-clock time per query for each stage of the pipeline on a single NVIDIA A100 GPU. The total runtime is dominated by the paraphrase generation and embedding extraction steps, which rely on LLM inference and scale with sequence length. In contrast, the specific components introduced by our method, which include training the proxy model and calculating the PA scores, add negligible overhead, requiring less than 0.2 seconds combined. Thus, the computational cost is primarily bound by the LLM's generation speed. For time-sensitive applications, this overhead can be significantly reduced by parallelizing the paraphrase generation or employing faster models.

## F.9 ALTERNATIVE POOLING AND LAYER CHOICES

To evaluate whether the PA nonconformity score depends on a particular LLM representation, we conduct a controlled sensitivity analysis on the `Qwen2.5-7B-Instruct` model using the **CosmosQA** benchmark. In the main paper, PA uses the last-layer hidden state at the final token position. Here, we consider three alternative representations that are commonly used in transformer analysis:

- **Mean pooling over tokens.** We average hidden states across all sequence positions (masking padding tokens). This yields a sequence-level representation that does not rely on a specific token location.

- **Intermediate-layer hidden states.** Instead of the last layer, we extract representations from a configurable layer index. In the experiment below, we uses the second-to-last layer). We keep the extraction location fixed (final token), so only the layer depth varies.

- **Attention-weighted pooling.** We use attention weights from the last layer to produce a weighted average over token embeddings. Specifically, we take the last token's attention distribution, average over attention heads, and use this vector as weights over the hidden states of all tokens.

**Results.** Tables 14 and 15 report coverage rate (CR) and set size (SS) under split CP. Across all pooling and layer choices, PA-based nonconformity scores remain extremely stable. This demonstrates that PA does *not* depend on a specific pooling strategy or layer depth, and that the semantic structure encoded throughout the transformer stack is sufficient for proxy training.

Table 12: **Paraphrase quality evaluation** using Llama-3.1-8B and Qwen2.5-7B as paraphrase generators. We report SBERT cosine similarity, BLEU, and ROUGE-L (mean $\pm$ std) for three paraphrase scenarios: $\mathcal{T}^*(x)$, $\mathcal{T}(x)$, and $\mathcal{T}^*(\mathcal{T}(x))$.

| RC (CosmosQA) | | | |
|---|---|---|---|
| **Scenario / Model** | **Cosine** | **BLEU** | **ROUGE-L** |
| $\mathcal{T}^*(x)$ — *paraphrases for human-written inputs* | | | |
| Llama | $0.82 \pm 0.10$ | $16.68 \pm 8.57$ | $45.18 \pm 12.02$ |
| Qwen | $0.75 \pm 0.11$ | $7.21 \pm 4.35$ | $26.86 \pm 8.35$ |
| $\mathcal{T}(x)$ — *LLM-reworded inputs* | | | |
| Llama | $0.70 \pm 0.21$ | $11.94 \pm 13.24$ | $35.84 \pm 19.98$ |
| Qwen | $0.72 \pm 0.20$ | $9.81 \pm 9.96$ | $32.05 \pm 16.85$ |
| $\mathcal{T}^*(\mathcal{T}(x))$ — *paraphrases of LLM-reworded inputs* | | | |
| Llama | $0.70 \pm 0.20$ | $11.73 \pm 9.83$ | $35.62 \pm 16.60$ |
| Qwen | $0.65 \pm 0.20$ | $6.53 \pm 5.92$ | $25.06 \pm 11.90$ |

| CI (HellaSwag) | | | |
|---|---|---|---|
| **Scenario / Model** | **Cosine** | **BLEU** | **ROUGE-L** |
| $\mathcal{T}^*(x)$ — *paraphrases for human-written inputs* | | | |
| Llama | $0.49 \pm 0.03$ | $9.80 \pm 2.08$ | $34.39 \pm 4.53$ |
| Qwen | $0.53 \pm 0.04$ | $11.50 \pm 3.59$ | $37.92 \pm 5.24$ |
| $\mathcal{T}(x)$ — *LLM-reworded inputs* | | | |
| Llama | $0.37 \pm 0.15$ | $6.71 \pm 4.64$ | $24.65 \pm 11.55$ |
| Qwen | $0.46 \pm 0.16$ | $6.98 \pm 9.41$ | $24.07 \pm 12.22$ |
| $\mathcal{T}^*(\mathcal{T}(x))$ — *paraphrases of LLM-reworded inputs* | | | |
| Llama | $0.37 \pm 0.13$ | $6.66 \pm 2.75$ | $24.61 \pm 8.13$ |
| Qwen | $0.43 \pm 0.15$ | $5.31 \pm 2.91$ | $24.61 \pm 6.92$ |

| MedMCQA | | | |
|---|---|---|---|
| **Scenario / Model** | **Cosine** | **BLEU** | **ROUGE-L** |
| $\mathcal{T}^*(x)$ — *paraphrases for human-written inputs* | | | |
| Llama | $0.81 \pm 0.10$ | $10.41 \pm 6.50$ | $38.25 \pm 11.39$ |
| Qwen | $0.80 \pm 0.09$ | $7.33 \pm 4.91$ | $31.13 \pm 9.35$ |
| $\mathcal{T}(x)$ — *LLM-reworded inputs* | | | |
| Llama | $0.61 \pm 0.27$ | $5.81 \pm 5.69$ | $24.51 \pm 14.60$ |
| Qwen | $0.65 \pm 0.29$ | $8.15 \pm 12.05$ | $28.74 \pm 21.24$ |
| $\mathcal{T}^*(\mathcal{T}(x))$ — *paraphrases of LLM-reworded inputs* | | | |
| Llama | $0.61 \pm 0.27$ | $5.81 \pm 5.69$ | $24.51 \pm 14.60$ |
| Qwen | $0.60 \pm 0.27$ | $4.11 \pm 4.11$ | $20.42 \pm 11.82$ |

Table 13: Runtime analysis (seconds per question) for $m = 6$ paraphrases

| Task | MMLU | CosmosQA | HellaSwag | HaluDial | HaluSum |
|---|---|---|---|---|---|
| Paraphrase generation | 4.55s | 2.74s | 2.92s | 2.03s | 1.93s |
| LLM embedding extraction | 1.81s | 0.619s | 3.59s | 0.651s | 4.67s |
| Proxy model training | 0.156s | 0.160s | 0.158s | 0.166s | 0.181s |
| PA score calculation | 0.00120s | 0.00502s | 0.00251s | 0.00108s | 0.00109s |

## F.10 SCORE ANALYSIS

We analyze the properties of the Mean, Weighted, and Worst nonconformity scores across the five general QA benchmarks, calculating statistics over the calibration and test splits. As shown in Table 16, the Worst score consistently exhibits a significantly higher mean and standard deviation

Table 14: Coverage Rate (CR, %) under different pooling and layer choices for `Qwen2.5-7B-Instruct` on CosmosQA. CR is better if closer to 90%. Results are reported for both the normal and fully reworded (Full) settings. Best results are in bold.

| CP Type | Method | Last Hidden (Paper) | | Attention Weighted | | Intermediate Layer | | Mean Pooling | |
|---|---|---|---|---|---|---|---|---|---|
| | | Normal | Full | Normal | Full | Normal | Full | Normal | Full |
| Split CP | LAC | 89.00 | 90.33 | 89.47 | 88.86 | **89.47** | 88.83 | **89.47** | 88.83 |
| | APS | 93.00 | 92.27 | 99.67 | 90.93 | 99.67 | 90.90 | 99.67 | 90.90 |
| | PA | **89.77** | **90.67** | **89.90** | **90.10** | 90.80 | **89.60** | 90.77 | **89.43** |
| QCCP | LAC | 88.87 | 90.63 | **90.00** | 89.20 | **90.00** | 89.20 | **90.00** | 89.20 |
| | APS | 93.00 | 92.50 | 98.67 | 91.06 | 97.30 | 91.03 | 97.30 | 91.03 |
| | PA | **90.10** | **90.33** | 89.73 | **90.07** | 90.60 | **89.50** | 90.97 | 89.17 |

Table 15: Prediction set size (SS, ↓) under different pooling and layer choices for `Qwen2.5-7B-Instruct` on CosmosQA. Lower SS indicates sharper uncertainty in Normal and fully reworded (Full) settings. Best results are in bold.

| CP Type | Method | Last Hidden (Paper) | | Attention Weighted | | Intermediate Layer | | Mean Pooling | |
|---|---|---|---|---|---|---|---|---|---|
| | | Normal | Full | Normal | Full | Normal | Full | Normal | Full |
| Split CP | LAC | 3.41 | 3.87 | **1.31** | 4.71 | 1.31 | 4.71 | 1.31 | 4.71 |
| | APS | 4.20 | 4.19 | 5.64 | 4.82 | 5.63 | 4.82 | 5.63 | 4.82 |
| | PA | **1.19** | **1.32** | 2.54 | **1.25** | **1.14** | 2.37 | **1.66** | **2.00** |
| QCCP | LAC | 3.40 | 3.91 | **1.68** | 4.74 | 1.68 | 4.74 | 1.68 | 4.74 |
| | APS | 4.20 | 4.22 | 4.65 | 4.83 | 3.65 | 4.83 | 3.65 | 4.83 |
| | PA | **1.21** | **1.30** | 2.53 | **1.25** | **1.18** | 2.44 | **1.67** | **2.01** |

compared to the Mean and Weighted variants. Since a higher nonconformity score corresponds to lower predicted probability (lower confidence), this disparity highlights the brittleness of LLM predictions: models often struggle with specific "hard" paraphrases, assigning high nonconformity scores to the correct answer even when they answer other paraphrases correctly.

In contrast, the Mean and Weighted scores remain lower and more stable (lower standard deviation). This confirms that our aggregation strategy effectively smooths out the noise induced by brittle paraphrases, resulting in uncertainty estimates that are more representative of the model's true semantic knowledge rather than its sensitivity to surface-level wording.

Table 16: Score statistics displayed as mean ± std for the correct answer choice on each dataset. These statistics are calculated using scores from both the calibration and test sets. A lower score indicates higher confidence (higher predicted probability) from the proxy model.

| Dataset | Mean Score | Weighted Score | Worst Score |
|---|---|---|---|
| MMLU | 0.427 ± 0.349 | 0.416 ± 0.364 | 0.713 ± 0.413 |
| CosmosQA | 0.229 ± 0.294 | 0.229 ± 0.319 | 0.483 ± 0.438 |
| HellaSwag | 0.262 ± 0.346 | 0.259 ± 0.354 | 0.436 ± 0.435 |
| HaluDial | 0.306 ± 0.343 | 0.304 ± 0.350 | 0.527 ± 0.435 |
| HaluSum | 0.119 ± 0.191 | 0.143 ± 0.235 | 0.315 ± 0.392 |

## F.11 EXTENSIONS TO OTHER TASKS

While our primary framework addresses tasks with finite label spaces (i.e., $|\mathcal{Y}| < \infty$), it naturally extends to critical sub-problems within open-ended generation, specifically *factuality verification*. Although the space of generated text is theoretically infinite, modern evaluation protocols typically decompose long-form generations into discrete atomic claims, each evaluated independently against a knowledge source (Mohri & Hashimoto, 2024; Cherian et al., 2024).

This decomposition effectively reduces the verification problem to a binary classification task. Formally, let an open-ended generation be parsed into a set of atomic claims $\{c_1, c_2, \ldots, c_k\}$. We treat each claim $c_i$ as an input $x \in \mathcal{X}$, where the label space is restricted to $\mathcal{Y} = \{\text{True}, \text{False}\}$.

Our paraphrase-robust framework is particularly well-suited for this setting due to the sensitivity of LLM-based evaluators to surface-level phrasing. To apply our method:

1. **Paraphrase Generation:** For a given claim $x$, we generate a set of semantic paraphrases $B(x)$.

2. **Proxy Scoring:** We utilize our lightweight proxy model $P_\theta(y \mid E(x))$ to predict the truthfulness of the claim based on the embeddings of these paraphrases.

3. **Robust Uncertainty:** We calculate PA scores across $B(x)$.

By integrating these scores into the conformal prediction framework, we ensure that the resulting uncertainty estimates, as well as the subsequent decision to flag a claim as hallucinated, are robust to lexical diversity. This guarantees that a claim is not penalized merely due to the specific syntax chosen by the LLM, but rather evaluated based on its semantic validity.

