# OpenReview forum: "Paraphrase-Robust Conformal Prediction for Reliable LLM Uncertainty Quantification"
_ICLR.cc/2026/Conference — Submitted to ICLR 2026_

### Official Review · Reviewer_9pty · 2025-10-19

**Soundness:** 2
**Presentation:** 3
**Contribution:** 2
**Rating:** 2
**Confidence:** 4

**Summary:**

This paper proposes a paraphrase-robust scoring function and evaluates it using existing conformal prediction frameworks on multiple-choice questions for LLMs. The framework involves training a lightweight classifier to achieve better model calibration and introducing paraphrased variations of the inputs to enhance robustness.

**Strengths:**

- Well-written paper with clear experimental demonstrations and ablations.
- Very important and timely problem.
- Released Code.

**Weaknesses:**

- The proposed approach builds entirely on existing conformal prediction frameworks without offering new theoretical insights.
- To preserve the theoretical guarantees of CP frameworks, the (X,Y) pairs must satisfy the exchangeability assumption, even under adversarial paraphrasing. This limitation weakens the contribution, as the proposed method effectively remains an application of CP frameworks when the test and calibration distributions are identical. I would expect a framework designed to maintain robustness under adversarial distribution shifts, which are typically unknown at calibration time.
 - As a suggestion: the adversarial paraphrasing for UQ methods has been discussed in early works: https://aclanthology.org/2025.acl-long.1429
- Overall, my concern is about the contribution of the paper. This paper combines existing ideas for LLMs in a convincing way, but I struggle to understand the unique perspective of the paper.

**Questions:**

- Do you consider any parahprashing as "adversarial"?

---

> ### Author Response · Authors · 2025-11-22
> **Response to to Reviewer 9pty (Part 1/4)**
>
> We thank the reviewer for the thoughtful review. We appreciate the recognition of the importance of the problem and the quality of our experiments and writing. Below we address each concern point-by-point.
>
> ---
>
> >**W1: The proposed approach builds entirely on existing conformal prediction frameworks without offering new theoretical insights.**
>
> We agree that our method builds on standard CP/QCCP and we do not claim to introduce a new CP framework. However, our contribution is more than an application: we provide new theoretical insights into *when and why* transformation-based, data-driven score design preserves CP validity.
>
> In the revision, we added a new **Theoretical Analysis** section (Sec. 3.3) that formalizes a general principle behind our approach:
> - We introduce **distributional closedness** (Assumption 3.1), which captures a broad class of meaning-preserving transformations—paraphrasing being a central and practically important instance.
> - Under this assumption, Proposition 3.1 proves that applying the same transformation $\mathcal{T}$ to all inputs **restores score-level exchangeability**, even when calibration and test sets differ in whether they were originally transformed. This provides a formal mechanism by which a semantic transformation can “repair’’ paraphrased-induced shift at the score level.
> - Lemma 3.1 further shows that our Paraphrase-Aware (PA) scores—deterministic, permutation-invariant functions of independently transformed inputs—**inherit this exchangeability** in the normal, fully-reworded and semi-reworded settings. As a result, CP/QCCP maintains their finite-sample coverage guarantees.
> These results characterize **when** a transformation-based nonconformity score (such as paraphrase aggregation) preserves CP validity and **why** it works. Although our empirical focus is on paraphrasing, the theory applies to any distributionally closed transformation (e.g., stylistic rewrites or synonym-based augmentations), offering a principled augmentation-based perspective on score design.
>
> To the best of our knowledge, this specific connection between distributionally closed semantic transformations, restored exchangeability, and valid CP for LLMs has not been made explicit in prior work. Reviewer `Kxds` also noted that this addresses a “genuine gap’’ in the current CP-for-LLM literature.
>
> > **W2: To preserve the theoretical guarantees of CP frameworks, the (X,Y) pairs must satisfy the exchangeability assumption, even under adversarial paraphrasing. This limitation weakens the contribution, as the proposed method effectively remains an application of CP frameworks when the test and calibration distributions are identical. I would expect a framework designed to maintain robustness under adversarial distribution shifts, which are typically unknown at calibration time.**
> Our goal is not to solve arbitrary, unknown distribution shifts, but to address a practically important and structured one: **meaning-preserving paraphrasing**. We clarify the logic here.
>
> 1. **PA does not assume knowledge of the test-time shift.**
> The proxy model and PA score are trained **only on the original calibration data**. In the semi-reworded setting, the calibration data remain clean, and only the test question is reworded by an LLM—this shift is *never* observed during training. PA does not adapt to the shift; instead, it applies the same paraphrasing-and-aggregation procedure to the test point at inference. This symmetric treatment restores **score-level exchangeability**, which is the only assumption CP requires.
> 2. **Why PA maintains exchangeability without seeing paraphrases during training.**
> Section 3.3 introduces the notion of **distributional closedness**, where the paraphrasing operator $\mathcal{T}$ satisfies $\mathcal{T}(X) \overset{d}{=} \mathcal{T}\circ \mathcal{T}(X)$. Under this condition, applying $\mathcal{T}$ to the test example but not to the calibration examples is score-equivalent to evaluating *all* points in the “once-transformed” domain. Because PA scores apply $\mathcal{T}$ identically to every generated paraphrase, Proposition 3.1 and Lemma 3.1 show that $S(\mathcal{T}(X_{n+1}), Y_{n+1})$ remains exchangeable with $\{S(X_i, Y_i)\}_{i=1}^n$  even though $\mathcal{T}(X_i)$ was never seen during training.
> In other words, **PA restores exchangeability—and thus CP validity—without requiring the shift to be known, observed, or adapted to during calibration**. The only requirement is the closedness property, which naturally holds for meaning-preserving paraphrasing and similar invariant rewrites.
>
> (Continue to next part)

---

> ### Author Response · Authors · 2025-11-22
> **Response to to Reviewer 9pty (Part 2/4)**
>
> (Continued from the previous part of response to W2)
>
> 3. **Empirical evidence:**
>
> We now explicitly evaluate this case through a new **semi-reworded setting**, where the training and calibration questions remain in their original human-written form and *only the test question is reworded by an LLM*. For PA score computation, we always paraphrase the input question; the only difference is whether the base input to the PA paraphraser is a human-written question or an LLM-generated rewrite (as illustrated in Fig. 1, where $Q$ is human-written and $Q'$ is an LLM paraphrase).
>
> In the newly added semi-reworded experiments—using either Qwen or Llama for the paraphrasing and rewording—PA scores maintain near-nominal coverage and achieve the smallest prediction-set sizes across RC, CI, and MedMCQA. The trends closely match those observed in the normal and fully-reworded settings, demonstrating that PA remains robust even when the paraphrase generator and the MCQA model are **independent** and the shift is entirely unseen during training. The first table presents the results obtained using Qwen, while the second table compares Qwen and Llama performance on the CosmosQA and HellaSwag datasets.
>
> | Method | QA (MMLU) | RC (CosmosQA) | CI (HellaSwag) | DRS (HaluDial) | DS (HaluSum) |
> | :--- | :---: | :---: | :---: | :---: | :---: |
> | **_Coverage Rate – CR (%) — Standard_** | | | | | |
> | LAC | 96.13 | 83.7 | 92.63 | 99.67 | **95.6** |
> | APS | 96.3 | **91.03** | 95.5 | 99.5 | 97.67 |
> | Paraphrase | **87.1** | 87.03 | **89.6** | **90.07** | 83.6 |
> | **_Coverage Rate – CR (%) — Quasi_** | | | | | |
> | LAC | 94.5 | 83.77 | 87.7 | 99.9 | **96.2** |
> | APS | 93.77 | **91** | 98.8 | 95.53 | 99.27 |
> | Paraphrase | **86.33** | 87.4 | **88.5** | **89.97** | 83.6 |
> | **_Prediction Uncertainty – SS $\downarrow$ — Standard_** | | | | | |
> | LAC | 5.51 | 3.12 | 4.46 | 5.93 | 4.36 |
> | APS | 5.69 | 4.02 | 4.93 | 5.89 | 4.69 |
> | Paraphrase | **2.73** | **1.21** | **2.22** | **2.27** | **2.06** |
> | **_Prediction Uncertainty – SS $\downarrow$ — Quasi_** | | | | | |
> | LAC | 5.35 | 3.11 | 3.56 | 5.98 | 4.45 |
> | APS | 5.52 | 4.02 | 5.74 | 5.19 | 5.6 |
> | Paraphrase | **2.71** | **1.23** | **2.17** | **2.28** | **2.06** |
>
>
> | | RC (CosmosQA) Qwen | RC (CosmosQA) Llama | CI (HellaSwag) Qwen | CI (HellaSwag) Llama |
> | :--- | :---: | :---: | :---: | :---: |
> | **Method** | | | | |
> | *Coverage Rate (CR, %) — Split CP* | | | | |
> | LAC | 83.70 | 87.17 | 92.63 | 86.97 |
> | APS | **91.03** | 99.33 | 95.50 | 90.90 |
> | PA | 87.03 | **89.97** | **89.60** | **90.27** |
> | *Coverage Rate (CR, %) — QCCP* | | | | |
> | LAC | 83.77 | 87.00 | 87.70 | 87.77 |
> | APS | **91.00** | 97.40 | 98.80 | 95.10 |
> | PA | 87.40 | **90.00** | **88.50** | **90.43** |
> | *Set Size (SS) ↓ — Split CP* | | | | |
> | LAC | 3.12 | 3.51 | 4.46 | 3.91 |
> | APS | 4.02 | 5.85 | 4.93 | 4.79 |
> | PA | **1.21** | **1.41** | **2.22** | **2.03** |
> | *Set Size (SS) ↓ — QCCP* | | | | |
> | LAC | 3.11 | 3.51 | 3.56 | 3.98 |
> | APS | 4.02 | 5.41 | 5.74 | 5.33 |
> | PA | **1.23** | **1.43** | **2.17** | **2.01** |

---

> ### Author Response · Authors · 2025-11-22
> **Response to to Reviewer 9pty (Part 3/4)**
>
> > **W3: As a suggestion: the adversarial paraphrasing for UQ methods has been discussed in early works: https://aclanthology.org/2025.acl-long.1429**
>
> We thank the reviewer for this pointer and have cited and discussed this work in the revised paper Appendix F.6. Conceptually, that line of work studies how paraphrasing affects **heuristic UQ metrics** (e.g., semantic-entropy-style scores) and typically does not provide distribution-free coverage guarantees. Our contribution is complementary: we study **conformal prediction with explicit coverage guarantees** and analyze how to design paraphrase-robust scores that remain valid under meaning-preserving variations. In particular, our new theory in Section 3.3 connects a semantic notion (paraphrasing as a distributionally closed transformation) to the core CP requirement of score exchangeability and shows how paraphrase aggregation can be used to recover coverage even under semantic shifts.
>
> ---
> > **W4. Overall, my concern is about the contribution of the paper. This paper combines existing ideas for LLMs in a convincing way, but I struggle to understand the unique perspective of the paper.**
>
> We respectfully clarify the originality of our contribution. To summarize our view:
> 1. **New empirical finding – paraphrase-sensitive CP failure.** We demonstrate that standard CP scores can produce 2-4× larger sets under meaning-preserving paraphrases even when exchangeability holds, a phenomenon not examined in prior CP-for-LLM work.
> 2. **Paraphrase-aware, data-driven scores with guarantees.** We propose PA scores that combine paraphrase augmentation and a learned proxy model, and we now prove conditions (Section 3.3) under which these scores preserve CP/QCCP coverage in normal, fully-reworded, and semi-reworded regimes.
> 3. **A general augmentation-based perspective on score design.** Our framework illustrates a broader principle: when we have access to a structured, closed-form transformation that models the shift (paraphrasing in our case), we can embed this transformation into the score design itself and still enjoy distribution-free CP guarantees.
>
> As shown in the table below, existing CP and UQ methods either (i) lack robustness to semantic paraphrasing or (ii) do not provide distribution-free coverage guarantees. Our work is the first to jointly offer **semantic robustness, theoretical coverage preservation under paraphrasing, and substantial practical gains in prediction-set efficiency**, positioning PA scores as a principled and effective score-design strategy for LLM uncertainty quantification.
>
> | Work  | Goal | Handles Semantic Paraphrasing? | Guarantees Provided | Score Design | CP Used? | Difference from Our Paper |
> |-----------------|------|--------------------------------|---------------------|--------------|----------|--------------------------------|
> | **Our paper (PA score)** | Paraphrase-robust CP for LLM MCQA | **Yes (explicitly modeled)** via distributionally-closed paraphrasing | **Valid marginal / quasi-conditional coverage** even under paraphrasing shift | **Data-driven paraphrase aggregation + proxy model** | Yes (CP + QCCP) | We design **semantic-robust scores**, provide **theory for exchangeability restoration**, and show **2–4× smaller sets** under paraphrasing. |
> | **LAC [1]** | CP for classification | No | Marginal coverage (standard CP) | Likelihood-based score | Yes | Sensitive to paraphrases |
> | **APS [2]** | Alternative CP score for classification | No | Marginal coverage | Adaptive prediction score | Yes | No robustness to semantic perturbations. |
> | **Semantic Entropy [3]** | UQ for LLMs via token entropy | **Heuristically robust** but no formal guarantees | None | Token-level entropy | No | No coverage guarantees; does not analyze exchangeability or paraphrasing rigorously. |
> | **Perturbation-based UQ (various)** | Improve robustness via input perturbation | Sometimes | None | Averaging or ensembling | No | No CP integration; no theoretical analysis of when perturbations preserve correctness. |
> | **Conformalized DL [4]** | Learn score functions via training objective | No | Marginal coverage | Learned score | Yes | Learns scores, but not designed for **semantic robustness** or paraphrase aggregation. |
> | **Boosted CP [5]** | Boost CP efficiency via boosting | No | Marginal coverage | Boosted features for scores | Yes | Improves efficiency, but does not consider semantic invariance or paraphrase-induced shifts. |
> | **CP with Learned Features [6]** | Learn representations for CP | No | Marginal coverage | Learned feature extractor | Yes | Trains features, but no semantic robustness focus. |
> | **Adversarial Paraphrase UQ [7])** | Stress-test UQ methods under paraphrasing | Yes (tests robustness) | None | Perturbation scoring | No | Does **not** address CP or provide guarantees. |
> | **LLM UQ Bench [8]** | Benchmark UQ metrics | Indirectly | None | Various heuristic scores | No | No CP, no score design targeted at semantic invariance. |

---

> ### Author Response · Authors · 2025-11-22
> **Response to to Reviewer 9pty (Part 4/4)**
>
> ---
> > **Q1: Do you consider any parahprashing as "adversarial"?**
>
> In our revised draft, we no longer refer to paraphrasing as “adversarial.” Our intention was not to treat *any* paraphrase as an adversarial attack. Instead, we study **meaning-preserving paraphrasing** generated by an LLM, which can nevertheless induce large changes in prediction-set size when using standard CP scores. In the revision, we use the terms **fully reworded** (both training/calibration and test inputs are LLM reworded questions) and **semi-reworded** (training/calibration inputs are human-written questions and test inputs are LLM reworded questions) to avoid confusion. Our setting is not adversarial in the usual sense; the paraphrases are semantic equivalents that should, ideally, not affect uncertainty estimates. Our goal is precisely to design scores that remain stable under such benign, semantically consistent variations.
>
> **Reference**
>
> [1] Sadinle, Mauricio, Jing Lei, and Larry Wasserman. "Least ambiguous set-valued classifiers with bounded error levels." Journal of the American Statistical Association 114.525 (2019): 223-234.
>
> [2] Romano, Yaniv, Matteo Sesia, and Emmanuel Candes. "Classification with valid and adaptive coverage." Advances in neural information processing systems 33 (2020): 3581-3591.
>
> [3] Kuhn, Lorenz, Yarin Gal, and Sebastian Farquhar. "Semantic uncertainty: Linguistic invariances for uncertainty estimation in natural language generation." arXiv preprint arXiv:2302.09664 (2023).
>
> [4] Einbinder, Bat-Sheva, et al. "Training uncertainty-aware classifiers with conformalized deep learning." Advances in neural information processing systems 35 (2022): 22380-22395.
> [5] Xie, Ran, Rina Barber, and Emmanuel Candes. "Boosted conformal prediction intervals." Advances in Neural Information Processing Systems 37 (2024): 71868-71899.
> [6] Kiyani, Shayan, George J. Pappas, and Hamed Hassani. "Conformal Prediction with Learned Features." International Conference on Machine Learning. PMLR, 2024.
>
> [7] Bakman, Yavuz Faruk, et al. "Reconsidering LLM uncertainty estimation methods in the wild." Proceedings of the 63rd Annual Meeting of the Association for Computational Linguistics (Volume 1: Long Papers). 2025.
>
> [8] Ye, Fanghua, et al. "Benchmarking llms via uncertainty quantification." Advances in Neural Information Processing Systems 37 (2024): 15356-15385.

---

> > ### Comment · Reviewer_9pty · 2025-11-26
> >
> > Thanks to the authors for their explanations and the modifications in the paper. With the current form, the paper is much better, but it's still almost a fully experimental paper. The theoretical insight is limited because the assumption (distributional assumption) is too strong. If we had a similar assumption for an operator, then the distribution shift wouldn't remain a problem in CP literature because we can transform the calibration distribution to the test distribution easily and make the problem iid. Still, I acknowledge the author's experimental contribution and raised my score accordingly.

---

> ### Author Response · Authors · 2025-11-27
>
> We thank Reviewer 9pty for the follow-up comment, for acknowledging the improvements in the revised manuscript, and for updating the score. We agree that the distributional-closedness assumption is non-trivial, and we do not intend it to serve as a general solution to arbitrary distribution shift. In the conformal prediction literature, it is well understood that distribution shift cannot be addressed without additional structural assumptions (e.g., invariances [1], covariate-label stability [2], or exchangeability-preserving transformations [3]).
>
> Our intention is more focused: to formalize a structured, meaning-preserving regime (paraphrasing), which is already implicitly assumed in many robustness and data-augmentation settings across NLP. In this restricted and practically important regime, the closedness property is natural: paraphrasing a paraphrase is designed to preserve semantic content and typically yields another draw from the same “semantic equivalence class”, a property broadly leveraged in paraphrase evaluation and semantic-invariance work.
>
> A promising possibility is to use the generative power of LLMs to learn or simulate distributional shifts directly. In principle, an LLM could be trained to mimic shifts in the data distribution, enabling us to design post-processing steps that maintain distributional closeness and score interchangeability. We see this direction as highly promising, but we leave it for future research, as realizing it fully will require substantial additional effort.
>
> Thank you again for your constructive feedback and for recognizing the experimental contribution of our work.
>
> **Reference**
>
> [1] Vovk, Vladimir, Alexander Gammerman, and Glenn Shafer. Algorithmic learning in a random world. Boston, MA: Springer US, 2005.
>
> [2]  Tibshirani, Ryan J., et al. "Conformal prediction under covariate shift." Advances in neural information processing systems 32 (2019).
>
> [3] Gibbs, Isaac, and Emmanuel Candes. "Adaptive conformal inference under distribution shift." Advances in Neural Information Processing Systems 34 (2021): 1660-1672.

---

### Official Review · Reviewer_iUNn · 2025-10-25

**Soundness:** 2
**Presentation:** 2
**Contribution:** 2
**Rating:** 4
**Confidence:** 5

**Summary:**

This paper aims to develop a paraphrase-robust uncertainty quantification framework based on split conformal prediction, which enhances the robustness of large language models under semantic variations by incorporating paraphrase-aware nonconformity scores. Experiments show that the method achieves smaller prediction sets and better stability across multiple MCQA datasets.

**Strengths:**

1. The paper presents the issue that when a query has multiple expressions, even though the semantics remain consistent, it can still affect the prediction set.

2. A probability vector predictor is trained using the hidden state, encoding and aggregating semantically consistent questions.

**Weaknesses:**

1. I believe that, at its core, the contribution of this paper is essentially the discovery that rephrasing, even when semantically equivalent, can impact the final prediction set size.

2. The reference to Figure 1 in the introduction is vague. From Figure 1 alone, I cannot discern the general work of the paper, nor can I see the comparison before and after paraphrasing. Additionally, the "two popular CP scores" are not clearly demonstrated.

3. I believe that since robustness is mentioned, the boundary of paraphrasing should be identified—specifically, when a problematic paraphrase occurs but still does not affect the prediction set.

Typo:
At the end of page five, the content should be appropriately adjusted. The citation of LLM-Uncertainty-Benchmark should not be added at the bottom of the page.

**Questions:**

1. The idea of PA is great. For example, when performing uncertainty decomposition, we also aggregate by rephrasing the question. Have you considered methods other than using the hidden state to train?

2. Does QCCP rely too much on "Conformal prediction with conditional guarantees"? I feel that typically, aiming for a marginal guarantee is sufficient, and there's no need to apply the conditional framework just to emphasize practical significance

3. Even if we don't rephrase the question, keeping calibration and test sets consistent, would there be a significant difference? For example, if a question in the test set is rephrased, would it break the exchangeability with the calibration data?

---

> ### Author Response · Authors · 2025-11-22
> **Response to to Reviewer iUNn (Part 1/4)**
>
> We thank the reviewer for the constructive comments. We appreciate your recognition for the importance of the problem and our method. We provide a point-to-point response below.
>
> ---
>
> > **W1. I believe that, at its core, the contribution of this paper is essentially the discovery that rephrasing, even when semantically equivalent, can impact the final prediction set size.**
>
> We thank the reviewer for this interpretation. We agree that identifying the effect of paraphrasing on prediction set size is an important empirical motivation, but this is only one part of our contribution. Our work goes substantially beyond observing this phenomenon.
>
> First, our analysis shows that the issue arises not simply from paraphrasing itself, but from a broader robustness problem: standard CP scores can be highly sensitive to surface-form variations in LLM outputs, even when semantic meaning and exchangeability are preserved. To address this, we develop a general theoretical framework (Sec. 3.3) based on **distributional closedness**, which characterizes when CP validity can be restored under controlled, meaning-preserving transformations. Although our experiments focus on paraphrasing—a practically important instance of this class—the theory applies to any such invariant transformation.
>
> Second, we introduce Paraphrase-Aware (PA) scores as a concrete instantiation of this framework. These scores are deterministic, permutation-invariant functions of independently transformed inputs, and the newly added Proposition 3.1 and Lemma 3.1 formally show that they restore score-level exchangeability under normal, fully-reworded, and semi-reworded settings. As a result, PA scores maintain CP/QCCP coverage while substantially reducing set size in our large-scale evaluation across datasets, LLMs, and paraphrase generators.
>
> In summary, while paraphrase-induced inflation of CP sets is an important empirical insight, the core contribution of our paper is broader: (i) revealing a general robustness issue for CP applied to LLMs, (ii) providing a theoretical principle for preserving validity under distributionally closed transformations, and (iii) designing data-driven nonconformity scores that leverage this principle to improve robustness without sacrificing coverage.
>
> ---
>
> > **W2. The reference to Figure 1 in the introduction is vague. From Figure 1 alone, I cannot discern the general work of the paper, nor can I see the comparison before and after paraphrasing. Additionally, the "two popular CP scores" are not clearly demonstrated.**
>
> Thank you for the helpful feedback. We have revised Figure 1 in the updated PDF to make the motivation and comparison much clearer.
>
> Specifically, the new Figure 1 now:
>
> - **Explicitly illustrates paraphrase sensitivity** of prior CP scores:
>   the original question and its LLM-rephrased variant produce *different option scores* and inflated prediction sets.
> - **Shows the before/after comparison side-by-side** so readers can visually see how set size expands under paraphrasing for baseline CP methods.
> - **Highlights that PA scores maintain efficiency** (smaller prediction sets) while stabilizing coverage across paraphrased inputs.
>
> We have also updated the introduction to explicitly reference Figure 1 as a motivating example of *paraphrase-induced instability* in existing CP scores and how our PA score resolves this issue.

---

> ### Author Response · Authors · 2025-11-22
> **Response to to Reviewer iUNn (Part 2/4)**
>
> > **W3. I believe that since robustness is mentioned, the boundary of paraphrasing should be identified—specifically, when a problematic paraphrase occurs but still does not affect the prediction set.**
>
>
> We appreciate this question. We agree that formally defining the “boundary of paraphrasing”,
> i.e., when a paraphrase remains meaning-preserving versus when it introduces a semantic change, is an interesting but currently open problem. To our knowledge, no prior CP-for-LLM work provides a precise or measurable definition of this boundary, and existing UQ papers typically rely on heuristic semantic-preservation assumptions.
>
> In our work, we therefore focus on **two practical and well-defined levels**:
>
> 1. **Theoretical side (distributional closedness).**
>  Our guarantees apply to any transformation $\mathcal{T}$ that is *distributionally closed*, meaning paraphrasing once or twice yields variants drawn from the same semantic distribution. This captures the notion of “meaning-preserving” without requiring a sharp boundary.
>
> 2. **Empirical side (semantic similarity).**
> We operationalize meaning preservation through explicit prompting and evaluate paraphrases using SBERT cosine similarity, BLEU, and ROUGE-L (See response to Reviewer Kxds W4 and Appendix F.7). Across datasets and two different paraphrase generators, the paraphrases consistently show moderate-to-high similarity, indicating that they remain within the semantic neighborhood where our assumptions hold.
>
> While quantifying the exact point at which a paraphrase becomes “problematic” is an important
> direction for future work, our empirical results across normal, fully-reworded, and semi-reworded settings suggest that PA scores remain stable across a broad range of natural paraphrasing styles.
>
> ---
>
> > **Typo: At the end of page five, the content should be appropriately adjusted. The citation of LLM-Uncertainty-Benchmark should not be added at the bottom of the page.**
>
> We have fixed this, thanks.

---

> ### Author Response · Authors · 2025-11-22
> **Response to to Reviewer iUNn (Part 3/4)**
>
> > **Q1. The idea of PA is great. For example, when performing uncertainty decomposition, we also aggregate by rephrasing the question. Have you considered methods other than using the hidden state to train?**
>
> Thank you for the suggestion. We have added several alternatives beyond the last-layer
> hidden state (LH). As shown in our new ablations below (detailed in Appendix F.9), we evaluate **attention-weighted pooling (AW)**, **intermediate-layer representations (IL)**, and **mean-pooled embeddings (MP)**, and find that PA scores remain stable across all of these choices. This indicates that our approach is not tied to a particular representation and that the proxy model benefits from the semantic structure encoded in many layers of the LLM.
>
> **Coverage Rate (CR %) for Qwen2.5-7B-Instruct on CosmosQA**
>
> CR Closer to 90% is better. Results are reported for both the normal and fully reworded (Full) settings. Best results are in bold.
>
> | CP Type  | Method | LH Normal | LH Full   | AW Normal | AW Full   | IL Normal | IL Full   | MP Normal | MP Full   |
> | -------- | ------ | --------- | --------- | --------- | --------- | --------- | --------- | --------- | --------- |
> | Split CP | LAC    | 89.00     | 90.33     | 89.47     | 88.86     | **89.47** | 88.83     | **89.47** | 88.83     |
> | Split CP | APS    | 93.00     | 92.27     | 99.67     | 90.93     | 99.67     | 90.90     | 99.67     | 90.90     |
> | Split CP | PA     | **89.77** | **90.67** | **89.90** | **90.10** | 90.80     | **89.60** | 90.77     | **89.43** |
> | QCCP     | LAC    | 88.87     | 90.63     | **90.00** | 89.20     | **90.00** | 89.20     | **90.00** | **89.20** |
> | QCCP     | APS    | 93.00     | 92.50     | 98.67     | 91.06     | 97.30     | 91.03     | 97.30     | 91.03     |
> | QCCP     | PA     | **90.10** | **90.33** | 89.73     | **90.07** | 90.60     | **89.50** | 90.97     | 89.17     |
>
>
> **Prediction Set Size (SS) for Qwen2.5-7B-Instruct on CosmosQA**
>
> SS Lower is better. Results are reported for both the normal and fully reworded (Full) settings. Best results are in bold.
>
>
> | CP Type  | Method | LH Normal | LH Full  | AW Normal | AW Full  | IL Normal | IL Full  | MP Normal | MP Full  |
> | -------- | ------ | --------- | -------- | --------- | -------- | --------- | -------- | --------- | -------- |
> | Split CP | LAC    | 3.41      | 3.87     | **1.31**  | 4.71     | 1.31      | 4.71     | 1.31      | 4.71     |
> | Split CP | APS    | 4.20      | 4.19     | 5.64      | 4.82     | 5.63      | 4.82     | 5.63      | 4.82     |
> | Split CP | PA     | **1.19**  | **1.32** | 2.54      | **1.25** | **1.14**  | **2.37** | **1.66**  | **2.00** |
> | QCCP     | LAC    | 3.40      | 3.91     | **1.68**      | 4.74     | 1.68      | 4.74     | 1.68      | 4.74     |
> | QCCP     | APS    | 4.20      | 4.22     | 4.65      | 4.83     | 3.65      | 4.83     | 3.65      | 4.83     |
> | QCCP     | PA     | **1.21**  | **1.30** | 2.53  | **1.25** | **1.18**  | **2.44** | **1.67**  | **2.01** |
>
>
>
>
> Other alternatives, such as using token-level features or attention maps, are promising directions, but we view them as orthogonal extensions. Our work focuses on demonstrating that a simple, frozen representation already provides robust paraphrase-aware scores.
>
> ---
>
> > **Q2. Does QCCP rely too much on "Conformal prediction with conditional guarantees"? I feel that typically, aiming for a marginal guarantee is sufficient, and there's no need to apply the conditional framework just to emphasize practical significance**
>
> We would like to clarify that **QCCP is not required for our method**, nor is our contribution dependent on conditional guarantees.
>
> - **PA scores work out-of-the-box with standard split CP.**
> Table 1 (split CP rows) already shows that our paraphrase-aware scores substantially reduce set size and improve robustness even under *pure marginal* CP. This demonstrates that our main contribution is the **score design**, not the calibration variant.
>
> - **QCCP is optional and orthogonal.**
> We include QCCP only as an additional refinement that can exploit heterogeneity across semantic clusters (e.g., different question types). It provides *finer-grained thresholds* but does not change the core algorithm.
>
> In summary, **our method does not rely on QCCP**. The core contribution, paraphrase-aware score design, remains fully effective under standard split CP, and QCCP is presented only as an optional enhancement for heterogeneous datasets.

---

> ### Author Response · Authors · 2025-11-22
> **Response to to Reviewer iUNn (Part 4/4)**
>
> > **Q3. Even if we don't rephrase the question, keeping calibration and test sets consistent, would there be a significant difference? For example, if a question in the test set is rephrased, would it break the exchangeability with the calibration data?**
>
> Thank you for raising this important question. We now explicitly evaluate this case through a new **semi-reworded setting**, where the calibration data remain in their original human-written form and **only the test question is reworded by an LLM**. For PA score computation, we always paraphrase the input question; the only difference is whether the base input to the paraphraser is a human-written question (calibration) or an LLM-reworded question (test). This is also illustrated in Fig. 1.
>
> Theoretically, Sec. 3.3 shows that this setting remains valid under our **distributional-closedness** assumption. Since the paraphrasing operator $\mathcal{T}$ satisfies
> $\mathcal{T}(X) \overset{d}{=} \mathcal{T}\circ \mathcal{T}(X)$,
> the PA score evaluated on the rephrased test input, $S(\mathcal{T}(X_{n+1}), Y_{n+1})$, is exchangeable with the PA scores on the calibration examples, $\{S(X_i, Y_i)\}_{i=1}^n$. Proposition 3.1 and Lemma 3.1 formally guarantee that score-level exchangeability holds, and therefore both CP and QCCP maintain their coverage guarantees in the semi-reworded scenario.
>
> Empirically, we report semi-reworded results using Qwen as both the QA model and paraphrase generator. Across all five datasets, PA scores continue to achieve near-nominal coverage and substantially smaller prediction sets than LAC and APS—mirroring the trends observed in the normal and fully-reworded settings. This confirms that rephrasing only the test question does **not** break validity for PA scores, while it does expose the robustness issues of baseline CP scores.
>
> | Method | QA (MMLU) | RC (CosmosQA) | CI (HellaSwag) | DRS (HaluDial) | DS (HaluSum) |
> | :--- | :---: | :---: | :---: | :---: | :---: |
> | **_Coverage Rate – CR (%) — Standard_** | | | | | |
> | LAC | 96.13 | 83.7 | 92.63 | 99.67 | **95.6** |
> | APS | 96.3 | **91.03** | 95.5 | 99.5 | 97.67 |
> | Paraphrase | **87.1** | 87.03 | **89.6** | **90.07** | 83.6 |
> | **_Coverage Rate – CR (%) — Quasi_** | | | | | |
> | LAC | 94.5 | 83.77 | 87.7 | 99.9 | **96.2** |
> | APS | 93.77 | **91** | 98.8 | 95.53 | 99.27 |
> | Paraphrase | **86.33** | 87.4 | **88.5** | **89.97** | 83.6 |
> | **_Prediction Uncertainty – SS $\downarrow$ — Standard_** | | | | | |
> | LAC | 5.51 | 3.12 | 4.46 | 5.93 | 4.36 |
> | APS | 5.69 | 4.02 | 4.93 | 5.89 | 4.69 |
> | Paraphrase | **2.73** | **1.21** | **2.22** | **2.27** | **2.06** |
> | **_Prediction Uncertainty – SS $\downarrow$ — Quasi_** | | | | | |
> | LAC | 5.35 | 3.11 | 3.56 | 5.98 | 4.45 |
> | APS | 5.52 | 4.02 | 5.74 | 5.19 | 5.6 |
> | Paraphrase | **2.71** | **1.23** | **2.17** | **2.28** | **2.06** |
>
>
>
>
>
>
>
>
> To address the concern about potential bias from using the same model for paraphrase generation and representation extraction, we additionally evaluate the semi-reworded setting using **Llama** as the paraphrase generator and **Qwen2.5-7B** as the MCQA model. As shown below, PA scores maintain near-nominal coverage and small set sizes—consistent with the trends in both the normal and fully reworded settings reported in our response to Reviewer Kxds (W5). This demonstrates that PA scores remain robust to paraphrase-induced variability even when the paraphrase generator and downstream LLM are completely independent. More details can be found in Appendix F.5.
>
> | | RC (CosmosQA)  Qwen | RC (CosmosQA)  Llama | CI (HellaSwag)  Qwen | CI (HellaSwag)  Llama |
> | :--- | :---: | :---: | :---: | :---: |
> | **Method** | | | | |
> | *Coverage Rate (CR, %) — Split CP* | | | | |
> | LAC | 83.70 | 87.17 | 92.63 | 86.97 |
> | APS | **91.03** | 99.33 | 95.50 | 90.90 |
> | PA | 87.03 | **89.97** | **89.60** | **90.27** |
> | *Coverage Rate (CR, %) — QCCP* | | | | |
> | LAC | 83.77 | 87.00 | 87.70 | 87.77 |
> | APS | **91.00** | 97.40 | 98.80 | 95.10 |
> | PA | 87.40 | **90.00** | **88.50** | **90.43** |
> | *Set Size (SS) ↓ — Split CP* | | | | |
> | LAC | 3.12 | 3.51 | 4.46 | 3.91 |
> | APS | 4.02 | 5.85 | 4.93 | 4.79 |
> | PA | **1.21** | **1.41** | **2.22** | **2.03** |
> | *Set Size (SS) ↓ — QCCP* | | | | |
> | LAC | 3.11 | 3.51 | 3.56 | 3.98 |
> | APS | 4.02 | 5.41 | 5.74 | 5.33 |
> | PA | **1.23** | **1.43** | **2.17** | **2.01** |

---

> > ### Comment · Reviewer_iUNn · 2025-11-23
> >
> > ok, thanks to authors for the supplemental explantions, I have raised my score, good luck. Also, the adversarial attack aspect represents a meaningful direction for deeper investigation in the future.

---

> > > ### Author Response · Authors · 2025-11-27
> > >
> > > We sincerely thank Reviewer iUNn for the thoughtful engagement throughout the discussion period and for the encouraging update to the score. We appreciate your recognition of the clarified explanation and your note regarding adversarial-style paraphrasing as a future research direction. We fully agree that exploring adversarial perturbations beyond meaning-preserving paraphrasing is a promising next step, and we have noted this as an avenue for future work in the last sentence of the conclusion section. Thank you again for your detailed reviews and constructive suggestions.

---

### Official Review · Reviewer_vAQA · 2025-10-26

**Soundness:** 3
**Presentation:** 3
**Contribution:** 2
**Rating:** 4
**Confidence:** 4

**Summary:**

Proposes a framework for conformal prediction in LLM MCQA under paraphrases by training a proxy MLP model on embeddings. The idea is that semantically similar phrases should give similar uncertainty estimates. Experiments on several QA datasets show tighter coverage and better robustness to adversarial paraphrasing under several LLM models.

**Strengths:**

* Clear problem and approach
* Well written and clear presentation
* Thorough evaluation on seven datasets and modern LLM with sufficient ablation experiments
* Identifies potential failure mode of CP in LLM
*

**Weaknesses:**

* Limited to QA classification tasks and not free-form generation
* Relies on embeddings and proxy model that might be unstable
* Dependency on paraphrase generator; a weak generator could lead of over-optimistic coverage
* The adversrial paraphrasing experiments could be more quantitative to validate robustness claims.
* Lack of runtime/cost analysis
* Proposed method is incremental
* Little theoretical analysis why paraphrase-aggregated scores preserved CP guarantees

**Questions:**

* How sensitive is performance to the quality or number of paraphrases?
* How to extend to other task such as summarization?
* What is overall overhead to inference in terms of wall time?
* How to ensure the paraphrase generator produces sufficiently diverse paraphrases that are still semantically related?
* What about using other embedding layers or different pooling aggregation such as attention weighted instead of mean pooling?
* Can you add confidence intervals or error bars to your plots?
* How does your method guarantee valid coverage across paraphrases? What about other forms of semantic shifts?
* How does the choice of paraphrase generator affect robustness?
* Why is the LLM projection head miscalibrated? Why does the proxy model fix this? Did you try calibrating the final layer with ECE loss?

---

> ### Author Response · Authors · 2025-11-22
> **Response to Reviewer vAQA (Part 1/5)**
>
> We thank the reviewer for the thoughtful and constructive feedback. We appreciate that you found the paper clear, well presented, and thoroughly evaluated. Below we address all concerns point-by-point.
>
> ---
>
> > **Summary: [...] The idea is that semantically similar phrases should give similar uncertainty estimates. [...]**
>
> We would like to clarify that the goal of our approach is not to train a model so that semantically similar phrases yield similar uncertainty estimates. Instead, our contribution is the design of paraphrase-robust nonconformity scores whose values remain stable even when the input question is reworded. This stability is what enables conformal prediction to achieve tighter prediction sets and near-nominal coverage under paraphrasing.
>
> ---
>
> > **W1. Limited to QA classification tasks and not free-form generation**
>
> Thank you for this comment. While our main experiments focus on QA-style classification, our method naturally extends to important subproblems in open-ended generation—most notably *factuality verification*, where long-form outputs are decomposed into atomic claims and evaluated as binary classification tasks. We provide detailed discussion in Appendix F.11; see also our response to Reviewer Kxds Q6 for how PA scores can be applied to factuality evaluation and open-ended generation more broadly.
>
>
> ---
>
> > **W2. Relies on embeddings and proxy model that might be unstable**
>
> Thank you for raising this concern. We have added analyses and ablations to clarify the stability of both the embedding representation and the proxy model.
>
> **1. Stability across embedding/pooling choices.**
> We evaluated several alternative representations beyond the last-layer last-hidden state used in the main paper, including: **attention-weighted pooling**,  **intermediate-layer representations**,  and **mean pooling**.
>
> Across all alternatives, PA scores consistently maintain approximately 90% coverage and produce compact prediction sets in both the normal and fully-reworded settings. A subset of results (Qwen2.5-7B) on three datasets is included below:
>
> | Setting | Representation | CR (Normal) | CR (Full-reword) | SS (Normal) | SS (Full-reword) |
> |--------|----------------|-------------|----------------|--------------|-----------------|
> | Split CP PA | Last hidden | 89.77 | 90.67 | 1.19 | 1.32 |
> |            | Attention-weighted | 89.90 | 90.10 | 2.54 | 1.25 |
> |            | Intermediate layer | 90.80 | 89.60 | 1.14 | 2.37 |
> |            | Mean pooling | 90.77 | 89.43 | 1.66 | 2.00 |
> | QCCP PA | Last hidden | 90.10 | 90.33 | 1.21 | 1.30 |
> |               | Attention-weighted | 89.73 | 90.07 | 2.53 | 1.25 |
> |               | Intermediate layer | 90.60 | 89.50 | 1.18 | 2.44 |
> |               | Mean pooling | 90.97 | 89.17 | 1.67 | 2.01 |
>
>
> **2. Proxy models are widely used and empirically stable.**
>
> The proxy model serves as a **learned nonconformity score**, a design choice that is consistent with prior work showing that learned scores can substantially improve CP efficiency while preserving coverage ([1], [2], [3]). These approaches all train a small model or feature extractor to produce more stable conformity statistics than raw model outputs.
>
> In our setting, the proxy plays the same conceptual role: it is a lightweight classifier trained on frozen LLM embeddings using a calibration-aware objective. This improves calibration and reduces variance when the question is rephrased by LLM (Figures 3 and 4), which in turn yields tighter CP prediction sets. Empirically, the proxy model yielded consistent results for 5 different random seeds as we added error bars to Figures 3, 4, 6, and 7.
>
> [1] Einbinder, Bat-Sheva, et al. "Training uncertainty-aware classifiers with conformalized deep learning." Advances in neural information processing systems 35 (2022): 22380-22395.
>
> [2] Xie, Ran, Rina Barber, and Emmanuel Candes. "Boosted conformal prediction intervals." Advances in Neural Information Processing Systems 37 (2024): 71868-71899.
>
> [3] Kiyani, Shayan, George J. Pappas, and Hamed Hassani. "Conformal Prediction with Learned Features." International Conference on Machine Learning. PMLR, 2024.

---

> ### Author Response · Authors · 2025-11-22
> **Response to Reviewer vAQA (Part 2/5)**
>
> > **W3. Dependency on paraphrase generator; a weak generator could lead of over-optimistic coverage**
>
> We thank the reviewer for raising this concern. We conducted additional experiments to directly evaluate the robustness of our method to the choice and quality of the paraphrase generator.
>
> **1. We test with *two independent paraphrase generators*.**
>
> In addition to Qwen2.5-7B, we regenerate all paraphrases using **Llama-3.1-8B** and report full results in our response to Reviewer Kxds (W5). The performance (in Appendix F.4) under Llama-generated paraphrases is **highly consistent** with Qwen-generated paraphrases (see RC, CI, and MedMCQA results), demonstrating that PA scores do not rely on a particular paraphrase generator.
>
> **2. We directly evaluate the semantic quality of the generated paraphrases.**
>
> As detailed in our response to Reviewer Kxds (W4), we measure paraphrase quality using SBERT cosine similarity, BLEU, and ROUGE-L across three datasets. The results (in Appendix F.7) show **moderate to high semantic similarity** for both generators (Qwen2.5-7B and Llama-3.1-8B), indicating that the paraphrases preserve the meaning of the original questions and are not overly “easy.”
>
> **3. Conformal coverage remains stable when paraphrase operation is distributionally closed.**
>
> Across Tables 1–3 in the main paper, both LAC and our PA score remain close to the nominal 90% coverage under the normal setting and fully reworded setting. This supports that conformal prediction maintains reliable coverage in our setting, and that PA scores do not artificially inflate performance.
>
>
> ---
>
> > **W4. The advers(a)rial paraphrasing experiments could be more quantitative to validate robustness claims.**
>
> Thank you for raising this point. To avoid confusion, we clarify that our paper does **not** use adversarial paraphrasing in the traditional sense (i.e., paraphrases intentionally crafted to break the model or violate semantic equivalence). Instead, all our experiments are built on **three controlled and theoretically motivated settings**, two of which satisfy the data exchangeability assumptions required for valid conformal prediction, and one of which is explicitly designed to evaluate robustness under rewording-induced shift. These settings align exactly with the theoretical scenarios analyzed in Proposition 3.1 and Lemma 3.1 in Section 3.3. We introduce the three settings briefly below.
>
> **1. Normal setting (no rewording-induced shift).**
> The training, calibration, and test sets all use the original human-written questions.
> This corresponds to the classic CP setting, where $S(X_{n+1},Y_{n+1}) \text{ is exchangeable with } \{S(X_i,Y_i)\}_{i=1}^n.$
>
> **2. Fully-reworded setting (rewording-induced shift applied uniformly for train/calibration and test set).**
> The training, calibration, and test sets **all receive the same transformation** via the paraphrase operator $\mathcal{T} $. In our experiment $\mathcal{T} $ is to use an LLM to reword human-written questions.
> Under this fully-reworded setup, we can restore exchangeability by applying the same distributionally closed operation to all data, as shown in Proposition 3.1.
> Formally,  $S(\mathcal{T}(X_{n+1}),Y_{n+1}) \text{ is exchangeable with }
> \{S(\mathcal{T}(X_i),Y_i)\}_{i=1}^n.$
>
> **3. Semi-reworded setting (test-time rewording-induced shiftt only).**
> In the semi-reworded setting, the calibration/training questions are human-written while the test question is rephrased by an LLM. Although this creates an input-level rewording-induced shiftt, our PA score computation removes this mismatch.
> Under the distributional closedness assumption (Assumption 3.1), applying the paraphrasing operator $\mathcal{T}$ twice yields a sample from the same distribution as applying $\mathcal{T}(X)$ once, so the score-level exchangeability still holds: $\bigl(\mathcal{T} \circ \mathcal{T}(X_{n+1}), Y_{n+1}\bigr)$ is exchangeable with
> {$(\mathcal{T}(X_i), Y_i)$}$_{i=1}^n$. Intuitively, while the original inputs differ (human-written vs. LLM-written), the PA scores are computed on paraphrased-once text for all examples: for calculating the PA score on the calibration set, we are using the LLM rewording of the human-written question, and for the test set PA score calculation, we use the LLM paraphrases of the LLM-reworded question, which is within the same distribution.
>
> This semi-reworded setting corresponds to the realistic case where the model is calibrated on original questions but encounters LLM produced rephrasings at test time. As clarified in our response to Reviewer iUNn (Q3), this is the setting most relevant for robustness. See Appendix F.5 for more details.

---

> ### Author Response · Authors · 2025-11-22
> **Response to Reviewer vAQA (Part 3/5)**
>
> > **W5. Lack of runtime/cost analysis.**
>
> We have added the wall time table for our pipeline to Appendix F.8. Each entry represents the average runtime per question.
>
> | Task | MMLU | CosmosQA | HellaSwag | HaluDial | HaluSum |
> | :--- | :---: | :---: | :---: | :---: | :---: |
> | Paraphrase generation | 4.55s | 2.74s | 2.92s | 2.03s | 1.93s |
> | LLM embedding extraction | 1.81s | 0.619s | 3.59s | 0.651s | 4.67s |
> | Proxy model training | 0.156s | 0.160s | 0.158s | 0.166s | 0.181s |
> | PA score calculation | 0.00120s | 0.00502s | 0.00251s | 0.00108s | 0.00109s |
>
> ---
>
> > **W6. “Proposed method is incremental.**
>
> We respectfully clarify that our contribution is not a new CP framework, but a **new robustness failure mode** and a **new score-design methodology** that directly addresses it. Both aspects are non-trivial and have not been explored in prior CP-for-LLM work.
>
> **1. We identify a previously overlooked failure mode of CP for LLMs.**
> Prior work (e.g., APS, LAC) implicitly assumes that CP scores remain stable under benign input rephrasings. We show—both theoretically and empirically—that *paraphrase sensitivity alone* can inflate prediction sets by **2–4×**, even when exchangeability holds. This reveals that CP can fail not due to distribution shift, but because LLM surface-form sensitivity disrupts the score structure itself.
>
> **2. We introduce a new class of *paraphrase-aware, data-driven nonconformity scores.**
> Score design is a central research direction in modern CP (e.g., Einbinder ’22; Kiyani ’24; Xie ’24). Our contribution fits into this line of work but is distinct in motivation and construction:
> - it aggregates **multiple paraphrases**, and
> - leverages a **learned proxy model**
> to produce **semantically robust** scores that suppress spurious surface-form variance.
> This yields **consistent reductions in set size**, while maintaining near-nominal coverage across datasets and LLMs.
>
> **3. Our findings highlight that score design within CP can yield additional robustness properties without sacrificing validity.**
> Our results show that CP validity remains unchanged—as guaranteed—but that the choice of nonconformity score can improve _semantic stability_, such as robustness to paraphrase variation. This demonstrates that optimized, paraphrase-aware score design can provide meaningful practical benefits on top of standard CP guarantees. We believe this perspective clarifies how CP can be adapted to LLM settings by focusing on learning scores that retain coverage while improving robustness and efficiency (or any other targeted properties).
>
> Taken together, our work provides:
> - a **new robustness insight** about CP for LLMs,
> - a **new paraphrase-aware nonconformity score design**, and
> - **comprehensive empirical validation** across multiple settings.
>
> We hope this clarifies that the contribution is not incremental, but instead offers **conceptual, methodological, and empirical advances** that strengthen CP for LLM uncertainty quantification.
>
> ---
>
> > **W7. Little theoretical analysis why paraphrase-aggregated scores preserved CP guarantees.**
>
> We address this concern in our new theoretical section (Sec. 3.3) and in our response to Reviewer Kxds (W1). In brief, CP requires only **score-level exchangeability**, not any specific functional form of the score. We prove that when paraphrasing is **distributionally closed**—that is, paraphrasing an already-paraphrased question yields another sample from the same distribution—then applying the *same paraphrasing-and-aggregation procedure* to every input **restores score-level exchangeability**, even when the original test and calibration inputs differ. This result is formalized in Proposition 3.1, which shows that our PA construction both preserves and slightly extends the conditions needed for valid CP.
>
> Our PA scores (mean, weighted, worst) are deterministic, permutation-invariant functions of the paraphrase set, which consists of independently generated paraphrased question variants. Because these aggregated scores apply the same transformation to every input, they **naturally inherit exchangeability across all three settings** we study (normal, fully-reworded, and semi-reworded), as formally proved in Lemma 3.1. Since both split CP and QCCP require only score-level exchangeability, their finite-sample coverage guarantees remain intact. This explains why paraphrase-aware aggregation can substantially improve empirical robustness while fully preserving the theoretical validity of conformal prediction.

---

> ### Author Response · Authors · 2025-11-22
> **Response to Reviewer vAQA (Part 4/5)**
>
> >**Q1. How sensitive is performance to the quality or number of paraphrases?**
>
> **Paraphrase quality.** Please see our response to Reviewer Kxds (W4): we conduct a detailed semantic-similarity analysis (SBERT cosine, BLEU, ROUGE-L) using two independent LLM paraphrase generators (Qwen and Llama). Across all datasets, paraphrases remain moderate-to-high cosine similarity, and PA scores achieve stable coverage for two different LLM paraphrase generators.
>
> **Number of paraphrases.**
> As shown in our ablation (Table 4), PA performance improves as the number of paraphrases increases. Coverage becomes more stable and set sizes decrease, indicating
> that aggregation benefits from more semantically-invariant samples.
>
>
> >**Q2.How to extend to other task such as summarization?**
>
> See our response to your W1 above.
>
>
> >**Q3. What is overall overhead to inference in terms of wall time?**
>
> See our response to your W5 above.
>
> >**Q4. How to ensure the paraphrase generator produces sufficiently diverse paraphrases that are still semantically related?**
>
> To encourage lexical and syntactic variation without altering semantics, we use a set of paraphrasing prompts that request different wording while explicitly instructing the model to preserve meaning. These prompts are included in our code release.
>
> As shown in our response to your W3 and to Reviewer Kxds (W4), we evaluate paraphrases using SBERT cosine similarity, BLEU, and ROUGE-L across three datasets. Both LLM paraphrase generators (Qwen2.5-7B and Llama-3.1-8B) produce paraphrases with **moderate-to-high semantic similarity**, confirming that they remain meaning-preserving.
>
>
>
> >**Q5. What about using other embedding layers or different pooling aggregation such as attention weighted instead of mean pooling?**
>
> See our response to your W2 where we added results for using three different alternative ways of pooling including attention weighted, intermediate-layer, and mean pooling.
>
> >**Q6. Can you add confidence intervals or error bars to your plots?**
>
> We have added error bars to all the plots and updated them in the PDF.
>
> >**Q7. How does your method guarantee valid coverage across paraphrases? What about other forms of semantic shifts?**
>
> Thank you for the question. Please see Sec. 3.3 and our response to W7 for the complete theoretical analysis. In brief, conformal prediction requires only **score-level exchangeability**, and our Paraphrase-Aware (PA) scores are explicitly constructed to satisfy this condition under a broad class of semantic shifts.
>
> Our theoretical results do **not** rely on paraphrasing per se. In Sec. 3.3, we introduce a general notion of **distributional closedness** (Asm. 3.1): a random transformation $\mathcal{T}$ such that $\mathcal{T}\circ\mathcal{T}(x)\overset{d}{=}\mathcal{T}(x)$ for any input or question $x$. Proposition 3.1 shows that under this assumption, applying the same transformation $\mathcal{T}$ to all inputs *restores exchangeability*, even if $\mathcal{T}$ introduces a meaning-preserving variation between calibration and test data. Lemma 3.1 then establishes that our PA scores (mean, weighted, worst)—being deterministic and permutation-invariant functions of independently sampled $\mathcal{T}(x)$—inherit this exchangeability in the normal, fully-reworded, and semi-reworded settings.
>
> Paraphrasing is one concrete instantiation of such a transformation: paraphrasing a paraphrase typically draws another sample from the same semantic-equivalence class. However, our theory applies equally to **any** semantic shifts that satisfy distributional closedness (e.g., stylistic rewrites and synonym substitutions, and other meaning-preserving invariances). In all such cases, split CP and QCCP retain their marginal or quasi-conditional coverage guarantees because the required symmetry holds at the score level.
>
> Shifts that alter the *semantic content* of the task falls outside the scope of standard conformal prediction and would require additional assumptions. Extending our approach to such semantic-altering shifts is an interesting direction for future work.

---

> ### Author Response · Authors · 2025-11-22
> **Response to Reviewer vAQA (Part 5/5)**
>
> > **Q8. How does the choice of paraphrase generator affect robustness?**
>
> We now consider Llama as our paraphrase generator on the CosmosQA, HellaSwag, and MedMCQA datasets and compare the results with Table 1. The performance is quite similar.
>
> (Fully reworded setting is abbreviated as Full)
> | Method | RC (Normal) | RC (Full) | CI (Normal) | CI (Full) | MedMCQA (Normal) | MedMCQA (Full) |
> | :--- | :--- | :--- | :--- | :--- | :--- | :--- |
> | **CR (%) — Split CP** | *(target 90%)* | | | | | |
> | LAC | 88.63 | 89.06 | **90.90** | **90.40** | **89.83** | **90.40** |
> | APS | 99.30 | 93.30 | 93.53 | 98.87 | 99.87 | 99.83 |
> | PA | **90.33** | **89.70** | 91.70 | 91.90 | 90.90 | 90.96 |
> | **CR (%) — QCCP** | *(target 90%)* | | | | | |
> | LAC | 89.03 | 89.03 | **91.06** | **91.27** | 96.83 | 91.17 |
> | APS | 94.90 | 97.23 | 97.30 | 95.17 | 96.13 | 97.00 |
> | PA | **90.23** | **89.80** | 92.13 | 91.90 | **90.70** | **91.03** |
> | **Set Size (SS) ↓ — Split CP** | | | | | | |
> | LAC | 2.81 | 3.69 | 3.37 | 4.23 | 3.57 | 5.03 |
> | APS | 5.77 | 4.50 | 4.66 | 5.88 | 5.98 | 5.98 |
> | PA | **1.16** | **1.30** | **1.70** | **1.96** | **2.55** | **2.98** |
> | **Set Size (SS) ↓ — QCCP** | | | | | | |
> | LAC | 2.84 | 3.76 | 3.39 | 4.42 | 5.19 | 5.10 |
> | APS | 4.44 | 5.34 | 5.34 | 5.29 | 5.10 | 5.66 |
> | PA | **1.15** | **1.49** | **1.72** | **2.22** | **2.56** | **3.10** |
>
>
>
> > **Q9. Why is the LLM projection head miscalibrated? Why does the proxy model fix this? Did you try calibrating the final layer with ECE loss?**
>
> The logits from the LLM projection head are optimized for **next-token prediction**, not MCQA uncertainty estimation, and are well-known to be miscalibrated and overconfident ([1], [2]). In Figure 3 of the paper, we compare the task performance with raw LLM logits and our proxy model trained on LLM hidden state. The proxy model, trained with a calibration-aware loss on frozen hidden states, achieves substantially lower Brier score and NLL.
>
> Importantly, **calibrating the LLM’s final layer is not a practical solution** in this setting:
> 1. It requires **task-specific finetuning** of a large model, which is expensive and must be repeated for every dataset and downstream task.
> 2. It also risks **interfering with next-token generation behavior**, as the projection head is shared across all tokens and tasks.
> 3. Our proxy is **lightweight, task-specific, plug-and-play**, and does not modify any part of the LLM. It can be trained cheaply on any dataset and reused across CP variants without touching the LLM.
>
> For these reasons, a small proxy model is a more stable and practical mechanism for producing
> calibrated, paraphrase-aware scores than repeatedly finetuning a large LLM head with ECE losses.
>
> [1] Zhou, Kaitlyn, Dan Jurafsky, and Tatsunori B. Hashimoto. "Navigating the grey area: How expressions of uncertainty and overconfidence affect language models." Proceedings of the 2023 Conference on Empirical Methods in Natural Language Processing. 2023.
>
> [2] Steyvers, Mark, et al. "What large language models know and what people think they know." Nature Machine Intelligence 7.2 (2025): 221-231.

---

### Official Review · Reviewer_Kxds · 2025-10-29

**Soundness:** 3
**Presentation:** 3
**Contribution:** 3
**Rating:** 6
**Confidence:** 4

**Summary:**

This paper addresses uncertainty quantification (UQ) for large language models (LLMs) by proposing a paraphrase-robust conformal prediction framework. The key insight is that existing conformal prediction (CP) methods for LLMs are sensitive to paraphrase variations, which can lead to unstable prediction sets. To address this, the authors introduce paraphrase-aware (PA) nonconformity scores that aggregate uncertainty across semantically equivalent rephrasings of each query. Experiments on five general MCQA datasets and two medical MCQA datasets with Qwen2.5-7B, Llama-3.1-8B, and Phi-3-small show that the method achieves nominal coverage (≈90%) with 2–4× smaller prediction sets than baselines (LAC, APS) under adversarial paraphrasing.

**Strengths:**

1. The paper identifies a genuine gap in existing conformal prediction for LLMs, which is lack of robustness to paraphrase variations. This is practical and relevant, as natural language queries can be expressed in many equivalent ways.
2. The use of paraphrase aggregation to achieve semantic invariance is intuitive and well-motivated. The three variants (mean, weighted, worst-case) provide a useful spectrum of robustness-efficiency trade-offs.
3. The datasets, LLM families and ablations in the experiments are extensive.
4. The method consistently achieves target coverage with substantially smaller prediction sets (often 2–4× reduction) compared to baselines, demonstrating practical value.

**Weaknesses:**

1. Why does paraphrase aggregation preserve/improve coverage? The paper does not provide theoretical analysis of how averaging scores across paraphrases affects the coverage guarantee. Under what conditions does this aggregation preserve the validity of conformal prediction?
2. While empirically robust, there are no formal guarantees (e.g., bounds on prediction set size variation) under paraphrase perturbations.
3. The connection between semantic invariance and statistical coverage is assumed but not rigorously established.
4. The paper does not verify that generated paraphrases truly preserve semantics. Are they evaluated by humans or using semantic similarity metrics?
5. Paraphrases are generated by an LLM (Qwen2.5-7B) and used to evaluate the same or similar LLMs. This could introduce systematic biases.
6. Generating 6 paraphrases per query increases computational cost by ≈6×. The paper does not report inference time or discuss computational efficiency.
7. The paper only compares against LAC and APS, which are relatively simple scores. Recent LLM-specific UQ methods (semantic entropy, perturbation-based methods mentioned in related work) were not considered as baselines.
8. Since paraphrases are generated from training/calibration samples and treated as additional samples, there may be data leakage or distribution shift issues not addressed.

**Questions:**

1. Can you provide theoretical analysis of coverage preservation under paraphrase aggregation? Specifically, under what conditions does averaging scores across paraphrases preserve the coverage guarantees of conformal prediction?
2. Can you prove or provide bounds on the coverage gap between S_mean(x,y) and S_prob(x,y)?
3. How do you ensure paraphrase quality and semantic preservation? Have you validated that generated paraphrases preserve semantics (human evaluation, semantic similarity scores)?
4. How does paraphrase quality affect the final results?
5. What is the computational cost and inference time? Can you report wall-clock time comparisons (with and without paraphrasing)?
6. Can you provide any preliminary results on open-ended generation, factuality verification, or other tasks mentioned in the related work?
7. Can you compare with more recent/relevant baselines mentioned above?
8. How does your method compare to simple ensembling or temperature-based uncertainty?
9. Since paraphrases of calibration samples are used in training the proxy, and paraphrases of test samples are used in evaluation, is there a risk of information leakage?
10. Have you tried generating paraphrases from a completely separate LLM to avoid circular dependencies?
11. Why does the "worst" score perform poorly (Figure 7)? Intuitively, the worst-case score should be most robust to adversarial paraphrasing, but it produces the largest sets and overshoots coverage. Can you explain this counterintuitive result?

---

> ### Author Response · Authors · 2025-11-22
> **Response to Reviewer Kxds (Part 1/5)**
>
> We sincerely thank the reviewer for the thoughtful and constructive feedback. We are glad that you found the paper’s motivation, method design, and empirical coverage-efficiency improvements meaningful. Below we address each concern point-by-point.
>
> ---
>
> > **W1. Why does paraphrase aggregation preserve/improve coverage? The paper does not provide theoretical analysis of how averaging scores across paraphrases affects the coverage guarantee. Under what conditions does this aggregation preserve the validity of conformal prediction?**
>
> Our paraphrase aggregation preserves conformal validity when the data remain exchangeable. We have added a new theoretical section (Sec. 3.3) to clarify this point. The core idea is that, although our score includes a random paraphrasing step, this step is applied through a fixed and independent procedure and therefore does not break the exchangeability already present in the data.
> Moreover, under a mild **distributionally closedness** assumption—namely, that paraphrasing an already-paraphrased sentence does not alter its distribution—the aggregation step can even restore exchangeability when paraphrasing introduces mean-preserving semantic shifts. Proposition 3.1 shows that applying the same transformation to all inputs realigns the calibration and test distributions, even if there exists a semantic variation caused by paraphrasing. Moreover, the added Lemma 3.1 establishes that all of our Paraphrase-Aware (PA) scores remain exchangeable in the normal, fully-reworded, and semi-reworded settings. Since score exchangeability is the only condition required for Split CP and QCCP to guarantee marginal or quasi-conditional coverage, these results formally justify the validity of our aggregation method.
>
>
> ---
>
> > **W2. While empirically robust, there are no formal guarantees (e.g., bounds on prediction set size variation) under paraphrase perturbations.**
>
> Section 3.3 now provides the formal conditions under which our PA scores preserve CP validity under paraphrase perturbations. While we do not bound the *size* of prediction-set variation—since it is generally hard to do so—our new results establish that, under the distributional-closedness assumption (Assumption 3.1), applying the same paraphrasing mechanism to calibration and test inputs restores exchangeability (Proproposition 3.1), even there exists a distributional shift cased by paraphrasing. Lemma 3.1 further shows that all aggregated PA scores remain exchangeable in normal, fully reworded, and semi-reworded settings. Thus, although set sizes may change with paraphrases, the **coverage guarantee** is theoretically preserved.
>
> ---
>
> > **W3. The connection between semantic invariance and statistical coverage is assumed but not rigorously established.**
>
> Section 3.3 makes explicit the connection between semantic invariance and statistical coverage. Distributional closedness formalizes the intuition that paraphrases preserve semantic content, and Proposition 3.1 proves that this property is sufficient to restore the exchangeability required for CP validity. Lemma 3.1 then shows that our PA scores inherit this exchangeability. This provides a rigorous justification for why semantic invariance via paraphrasing leads to preserved conformal coverage.

---

> ### Author Response · Authors · 2025-11-22
> **Response to Reviewer Kxds (Part 2/5)**
>
> > **W4. The paper does not verify that generated paraphrases truly preserve semantics. Are they evaluated by humans or using semantic similarity metrics?**
>
> Thank you for pointing this out. We have now added a quantitative evaluation of paraphrase quality across three datasets (CosmosQA, HellaSwag, MedMCQA). Specifically, we compute **SBERT cosine similarity**, **BLEU**, and **ROUGE-L** between the original questions and their paraphrases. We conduct this analysis using two independent paraphrase generators (**Llama-3.1-8B** and **Qwen2.5-7B**) and report metrics for all settings used in the paper.
>
> Before presenting the results, we first introduce the notation we use for paraphrase generators. Given a human-written question $x$, we use $\mathcal{T}$ to denote a single LLM-generated rewording of the input question,and we use $\mathcal{T}^*$ to denote the operation that generates a set of LLM-based paraphrases for any input (either human-written or LLM-reworded). Hence, there are three sets of paraphrased questions whose quality we evaluate:
>
> - $\mathcal{T}^*(x)$: six LLM-generated paraphrases per human-written question, used for PA score computation in the normal setting;
> - $\mathcal{T}(x)$: a single LLM-reworded question used as the test input in the fully reworded setting;
> - $\mathcal{T}^\*(\mathcal{T}(x))$: six LLM-generated paraphrases of the LLM-reworded question, used for PA score computation in the fully reworded setting.
>
> Across datasets, we observe **moderate to high SBERT cosine similarity** (0.70–0.82 for CosmosQA and MedMCQA; 0.37–0.53 for HellaSwag, reflecting its long-context nature), consistent BLEU scores, and stable ROUGE-L values. These results indicate that the paraphrases largely preserve the meaning of the original questions, despite natural stylistic variation. We added this analysis in Appendix F.7.
>
>
> **CosmosQA (Reading Comprehension) (10,000 questions)**
>
> | Scenario | Model | Cosine | BLEU | ROUGE-L |
> |---------|--------|--------|-------|----------|
> | $\mathcal{T}^*(x)$ | Llama |0.82 ± 0.10 | 16.68 ± 8.57 | 45.18 ± 12.02 |
> |  | Qwen | 0.75 ± 0.11 | 7.21 ± 4.35 | 26.86 ± 8.35 |
> | $\mathcal{T}(x)$ | Llama | 0.70 ± 0.21 | 11.94 ± 13.24 | 35.84 ± 19.98 |
> |  | Qwen | 0.72 ± 0.20 | 9.81 ± 9.96 | 32.05 ± 16.85 |
> | $\mathcal{T}^*(\mathcal{T}(x))$ | Llama | 0.70 ± 0.20 | 11.73 ± 9.83 | 35.62 ± 16.60 |
> |  | Qwen | 0.65 ± 0.20 | 6.53 ± 5.92 | 25.06 ± 11.90 |
>
> **HellaSwag (Commonsense Inference)  (10,000 questions)**
>
> | Scenario | Model | Cosine | BLEU | ROUGE-L |
> |---------|--------|--------|-------|----------|
> | $\mathcal{T}^*(x)$ | Llama | 0.49 ± 0.03 | 9.80 ± 2.08 | 34.39 ± 4.53 |
> |  | Qwen | 0.53 ± 0.04 | 11.50 ± 3.59 | 37.92 ± 5.24 |
> | $\mathcal{T}(x)$ | Llama | 0.37 ± 0.15 | 6.71 ± 4.64 | 24.65 ± 11.55 |
> |  | Qwen | 0.46 ± 0.16 | 6.98 ± 9.41 | 24.07 ± 12.22 |
> | $\mathcal{T}^*(\mathcal{T}(x))$ | Llama | 0.37 ± 0.13 | 6.66 ± 2.75 | 24.61 ± 8.13 |
> |  | Qwen | 0.43 ± 0.15 | 5.31 ± 2.91 | 24.61 ± 6.92 |
>
> **MedMCQA (Medical QA) (10,000 questions)**
>
> | Scenario | Model | Cosine | BLEU | ROUGE-L |
> |---------|--------|--------|-------|----------|
> | $\mathcal{T}^*(x)$ | Llama | 0.81 ± 0.10 | 10.41 ± 6.50 | 38.25 ± 11.39 |
> |  | Qwen | 0.80 ± 0.09 | 7.33 ± 4.91 | 31.13 ± 9.35 |
> | $\mathcal{T}(x)$ | Llama | 0.61 ± 0.27 | 5.81 ± 5.69 | 24.51 ± 14.60 |
> |  | Qwen | 0.65 ± 0.29 | 8.15 ± 12.05 | 28.74 ± 21.24 |
> | $\mathcal{T}^*(\mathcal{T}(x))$ | Llama | 0.61 ± 0.27 | 5.81 ± 5.69 | 24.51 ± 14.60 |
> |  | Qwen | 0.60 ± 0.27 | 4.11 ± 4.11 | 20.42 ± 11.82 |

---

> ### Author Response · Authors · 2025-11-22
> **Response to Reviewer Kxds (Part 3/5)**
>
> > **W5. Paraphrases are generated by an LLM (Qwen2.5-7B) and used to evaluate the same or similar LLMs. This could introduce systematic biases.**
>
> We thank the reviewer for bringing up this point. We have generated the paraphrases for CosmosQA, HellaSwag, and MedMCQA using a different LLM (Llama-3.1-8B) and compared the performance with the Qwen-generated paraphrases (see table below). We find that our results are quite consistent with Table 1, suggesting that any bias resulting from using the same LLM for paraphrasing and question-answering may be negligible.
>
> (Fully reworded is abbreviated as Full)
> | Method | RC (Normal) | RC (Full) | CI (Normal) | CI (Full) | MedMCQA (Normal) | MedMCQA (Full) |
> | :--- | :--- | :--- | :--- | :--- | :--- | :--- |
> | **CR (%) — Split CP** | *(target 90%)* | | | | | |
> | LAC | 88.63 | 89.06 | **90.90** | **90.40** | **89.83** | **90.40** |
> | APS | 99.30 | 93.30 | 93.53 | 98.87 | 99.87 | 99.83 |
> | PA | **90.33** | **89.70** | 91.70 | 91.90 | 90.90 | 90.96 |
> | **CR (%) — QCCP** | *(target 90%)* | | | | | |
> | LAC | 89.03 | 89.03 | **91.06** | **91.27** | 96.83 | 91.17 |
> | APS | 94.90 | 97.23 | 97.30 | 95.17 | 96.13 | 97.00 |
> | PA | **90.23** | **89.80** | 92.13 | 91.90 | **90.70** | **91.03** |
> | **Set Size (SS) ↓ — Split CP** | | | | | | |
> | LAC | 2.81 | 3.69 | 3.37 | 4.23 | 3.57 | 5.03 |
> | APS | 5.77 | 4.50 | 4.66 | 5.88 | 5.98 | 5.98 |
> | PA | **1.16** | **1.30** | **1.70** | **1.96** | **2.55** | **2.98** |
> | **Set Size (SS) ↓ — QCCP** | | | | | | |
> | LAC | 2.84 | 3.76 | 3.39 | 4.42 | 5.19 | 5.10 |
> | APS | 4.44 | 5.34 | 5.34 | 5.29 | 5.10 | 5.66 |
> | PA | **1.15** | **1.49** | **1.72** | **2.22** | **2.56** | **3.10** |
>
> ---
>
> > **W6. Generating 6 paraphrases per query increases computational cost by ≈6×. The paper does not report inference time or discuss computational efficient**
>
> We have added a full compute cost table. In summary:
>
> - Generating 6 paraphrases increases preprocessing cost by ~5.8×, but
> - **Prediction-time overhead is minimal** because the proxy is a small 2-layer MLP.
> - Our results in Table 4 of the paper show that only $m = 2$ paraphrases can still lead to a significant reduction in set size
>
> | Task | MMLU | CosmosQA | HellaSwag | HaluDial | HaluSum |
> | :--- | :---: | :---: | :---: | :---: | :---: |
> | Paraphrase generation | 4.55s | 2.74s | 2.92s | 2.03s | 1.93s |
> | LLM embedding extraction | 1.81s | 0.619s | 3.59s | 0.651s | 4.67s |
> | Proxy model training | 0.156s | 0.160s | 0.158s | 0.166s | 0.181s |
> | PA score calculation | 0.00120s | 0.00502s | 0.00251s | 0.00108s | 0.00109s |
>
> ---
>
> > **W7. The paper only compares against LAC and APS, which are relatively simple scores. Recent LLM-specific UQ methods (semantic entropy, perturbation-based methods mentioned in related work) were not considered as baselines.**
>
> We thank the reviewer for raising this point. We realize that the distinction between our setting and heuristic UQ methods was not made sufficiently explicit.
> Our work is entirely framed in **conformal prediction (CP)** for multiple-choice QA, where uncertainty is represented as a **per-option nonconformity score** used to construct **set-valued predictions with distribution-free coverage guarantees**. In contrast, semantic entropy and perturbation-based UQ methods provide **a single instance-level scalar uncertainty score** and are typically used for **hallucination detection or selective prediction**, without producing prediction sets or offering distribution-free coverage guarantees. Because the output objects and evaluation metrics are fundamentally different (prediction sets with coverage and set size vs. scalar scores with AUC/ECE), a direct numerical comparison would not be meaningful and would not answer the CP question we address.
>
> Conceptually, our approach is more closely related to recent work on **data-driven or learned nonconformity scores within CP**, such as conformalized deep learning and boosted CP intervals ([1], [2]) and conformal prediction with learned features ([3]). These methods, like ours, learn a score in order to improve CP efficiency while maintaining coverage guarantees, but they do not address paraphrase robustness or quasi-conditional guarantees for LLMs. We will revise the related work section to clarify this connection and to more clearly situate our contribution as a _learned, paraphrase-aware nonconformity score for CP_, rather than as another heuristic UQ measure.
>
> Finally, we already compare against the two strongest CP baselines widely used in the LLM UQ literature (LAC and APS), and we show that our learned score yields prediction sets with comparable coverage and substantially reduced set size, especially under paraphrase shifts. We hope this clarifies why we focus our empirical comparisons on CP-based methods rather than on non-CP UQ heuristics.

---

> ### Author Response · Authors · 2025-11-22
> **Response to Reviewer Kxds (Part 4/5)**
>
> > **W8. Since paraphrases are generated from training/calibration samples and treated as additional samples, there may be data leakage or distribution shift issues not addressed.**
>
> Thank you for raising this concern. We believe there may be a misunderstanding about how paraphrases are used in our framework. **Paraphrases are never added as extra training or calibration samples**, nor do they increase the number of data points seen by CP. Across all three evaluation settings, the number of input questions remains exactly the same as in the original dataset.
>
> Paraphrases are only used **internally to compute the PA score**, i.e., they appear inside the nonconformity score function for a single example. They are **not appended to the dataset**, not included in the calibration set, and not treated as independent observations. Thus, there is no risk of data leakage.
>
> Concretely, in the input-level transformation (Normal / Fully-reworded / Semi-reworded settings), we sometimes reword the input question using an LLM to study robustness. This changes the surface form of the question but does not add new samples.
>
> In contrast, for PA score computation, we internally generate 6 paraphrases of each question solely to compute its nonconformity score. These paraphrases are never used as question-level inputs for CP or QCCP.
>
> To avoid confusion, we have updated the manuscript to distinguish these two operations:
> - “Reword input question”: transformation applied at the dataset/input level (affects what CP sees as $X_i$)
> - “Generating paraphrases”: internal stochastic operation used only inside the PA score formula (does not alter the dataset)
>
> Finally, our theoretical analysis (Proposition 3.1 and Lemma 3.1) formalizes that these internal paraphrases preserve score-level exchangeability, ensuring CP validity even under rewording-induced shifts. This is also empirically confirmed across Normal, Fully-reworded, and Semi-reworded settings.
>
>
> ---
> ## Questions
>
> > **Q1. Can you provide theoretical analysis of coverage preservation under paraphrase aggregation? Specifically, under what conditions does averaging scores across paraphrases preserve the coverage guarantees of conformal prediction?**
>
> See our response for W1, W2, and W3 above.
>
> ---
>
> >**Q2. Can you prove or provide bounds on the coverage gap between S_mean(x,y) and S_prob(x,y)?**
>
> Thank you for the thoughtful question. As discussed earlier, both $S_{\text{mean}}$ and $S_{\text{prob}}$ lead to conformal predictors that enjoy the same finite-sample marginal coverage guarantee under our exchangeability assumptions. In particular, in the standard split conformal framework, the distribution of the empirical coverage depends only on the calibration sample size and the target miscoverage level, and does not depend on the specific choice of the nonconformity score. This implies that there is no systematic coverage gap between using $S_{\text{mean}}$ and $S_{\text{prob}}$; any small differences observed in practice are due to random fluctuations from a finite test set rather than an inherent deficiency of one score.
>
> For this reason, we do not expect a meaningful theoretical bound that separates the coverage of the two methods beyond the usual finite-sample fluctuation bounds already known for split conformal prediction. Instead, the main distinction between $S_{\text{mean}}$ and $S_{\text{prob}}$ lies in robustness: $S_{\text{mean}}$ aggregates multiple independent paraphrases and can yield tighter prediction sets in our experiments, whereas $S_{\text{prob}}$ uses only a single sample and is computationally cheaper. We therefore focus on comparing them empirically in terms of coverage and robustness, while maintaining the same theoretical coverage guarantee.
>
> We also offer an explanation for why the two approaches (our methods using $S_{\text{mean}}$ and LSA using $S_{\text{prob}}$) differ empirically. Our method uses a proxy model trained on an independent i.i.d. dataset to estimate prediction probabilities, whereas the baseline constructs prediction sets directly from the original model’s probability. Because our proxy model learns underlying patterns from additional data, it reduces randomness in the scoring process, which in turn leads to more stable estimates and therefore smaller prediction sets at the same coverage level.
>
> ---
>
> >**Q3. How do you ensure paraphrase quality and semantic preservation? Have you validated that generated paraphrases preserve semantics (human evaluation, semantic similarity scores)?**
>
> See our response for W4 above.
>
> ---
>
> >**Q4. How does paraphrase quality affect the final results?**
>
> See our response for W4 and W5 above. Although Llama slightly outperforms Qwen as a paraphrase generator in terms of the metrics described in our response for W4, the coverage and set size results are similar.

---

> ### Author Response · Authors · 2025-11-22
> **Response to Reviewer Kxds (Part 5/5)**
>
> >**Q5. What is the computational cost and inference time? Can you report wall-clock time comparisons (with and without paraphrasing)?**
>
> See our response to W6 for detailed timing results. Once the PA score is computed, the subsequent CP procedure is identical to that used for LAC and APS, so no additional overhead is introduced at inference time. As evidenced by the timing table, the extra cost comes almost entirely from paraphrase generation. The PA score computation and proxy model inference are effectively negligible, making our approach comparably efficient at prediction time to LAC and APS.
>
>
> ---
>
> >**Q6. Can you provide any preliminary results on open-ended generation, factuality verification, or other tasks mentioned in the related work?**
>
> We thank the reviewer for this suggestion. Our work focuses on demonstrating the effectiveness of PA scores on MCQA benchmarks, and applying our framework directly to open-ended generation is challenging due to the **infinite label space**. However, we agree that factuality verification is a promising direction for future work. Following the established literature (e.g., [4]), open-ended outputs can be decomposed into atomic claims, each of which can be evaluated as a binary classification problem ($\mathcal{Y} = {\text{True}, \text{False}}$). This conversion yields a finite label space compatible with our framework.
> Once the task is framed this way, our approach applies naturally: by training the proxy model on paraphrases of each claim, our PA scores can promote robust factuality judgments that are stable under lexical variation—precisely the weakness that motivates many recent verification methods.
>
> ---
>
> >**Q7. Can you compare with more recent/relevant baselines mentioned above?**
>
> See our justification for our choice of baseline in reply to W7 above.
>
> ---
>
> >**Q8. How does your method compare to simple ensembling or temperature-based uncertainty?**
>
> See our clarification for how our method is different from the heuristic uncertainty quantification method in W7 above.
>
> ---
>
> >**Q9. Since paraphrases of calibration samples are used in training the proxy, and paraphrases of test samples are used in evaluation, is there a risk of information leakage?**
>
> There is no risk of information leakage. The calibration set and test set are mutually exclusive. Paraphrases of calibration examples are only used to compute the score of their own parent sample, and are never treated as additional calibration items. CP calibration uses the **original calibration pairs**. Thus, exchangeability and marginal validity remain intact.
>
>
>
> ---
>
> >**Q10. Have you tried generating paraphrases from a completely separate LLM to avoid circular dependencies?**
>
> See our response for your W5. We added experiments paraphrasing with Llama-3.1-8B instead of Qwen2.5-7B. The results remain stable, indicating the method is not tied to any specific generator.
>
> ---
>
> >**Q11. Why does the "worst" score perform poorly (Figure 7)? Intuitively, the worst-case score should be most robust to adversarial paraphrasing, but it produces the largest sets and overshoots coverage. Can you explain this counterintuitive result?**
>
> We appreciate this insightful observation. We have updated the paper to clarify the limitations of the worst-case score: it is an inherently conservative measure due to its sensitivity to outliers (e.g., a single low-quality paraphrase). This sensitivity induces a heavy right tail in the score distribution, producing conservative thresholds that inflate prediction sets and overshoot coverage. This behavior is expected for max-based aggregations and is supported by the score statistics now included in Appendix F.10.
>
> [1] Einbinder, Bat-Sheva, et al. "Training uncertainty-aware classifiers with conformalized deep learning." Advances in neural information processing systems 35 (2022): 22380-22395.
>
> [2] Xie, Ran, Rina Barber, and Emmanuel Candes. "Boosted conformal prediction intervals." Advances in Neural Information Processing Systems 37 (2024): 71868-71899.
>
> [3] Kiyani, Shayan, George J. Pappas, and Hamed Hassani. "Conformal Prediction with Learned Features." International Conference on Machine Learning. PMLR, 2024.
>
> [4] Mohri, Christopher, and Tatsunori Hashimoto. "Language models with conformal factuality guarantees." Proceedings of the 41st International Conference on Machine Learning. 2024.

---

> ### Comment · Reviewer_Kxds · 2025-11-28
>
> I thank authors for their thorough responses to my questions, which have helped with my understanding of their core contributions. I don't have further clarification questions.

---

### Author Response · Authors · 2025-11-22
**Overall Response (Part 1/2)**

We extend our sincere gratitude for your thoughtful and constructive evaluations. Reviewers noted several strengths of the paper. Reviewer **Kxds** emphasized the practical importance of addressing paraphrase sensitivity in LLM uncertainty quantification, the clarity of our motivation, and the value of the three PA score variants. Reviewers **Kxds** and **vAQA** both highlighted the breadth of our empirical evaluation across seven datasets and multiple LLM families and found the problem setup clear and well articulated. Reviewer **iUNn** recognized the relevance of studying how semantically equivalent expressions can affect prediction-set size and noted that our experiments identify a real failure mode of conformal prediction in LLMs. Reviewer **9pty** acknowledged the importance and timeliness of the problem and the clarity of our experimental demonstrations. Across reviews, it was also noted that our method consistently achieves near-nominal coverage with substantial reductions in prediction-set size compared to standard CP scores.

## Clarification of Contribution
Several reviews (Kxds, vAQA, iUNn, 9pty) raised a central question: _what exactly is the contribution of this paper?_ We recognize that our original framing did not sufficiently separate (i) the classical conformal prediction (CP) framework from (ii) the part that is new in our work. In brief, our contribution is **a data-driven, paraphrase-robust design of nonconformity scores for LLM uncertainty quantification**. It shows how optimization and semantic structure can be used to construct scores that remain stable under meaning-preserving rewordings. We detail our contributions and answer shared questions below.


## Why does score design matter in conformal prediction for LLMs?

Classical CP theory focuses on providing distribution-free coverage guarantees **for any fixed score** under mild assumptions. As a result, prior work typically **treats the score as given** and does not investigate what makes a score good or robust in practice. However, our results (Table 1) show that existing score designs break down for LLMs: **conventional scores are highly sensitive to input-level paraphrasing**. When prompts undergo meaning-preserving rewrites, standard scores can produce massive and unpredictable inflation in prediction set size, even though the rewordings themselves do not alter the meaning of the prompt.
This last point is crucial. In our “fully reworded” setup (which was referred to as the ``adversarial setting’’ previously) , we paraphrase both the calibration data and the test data in the same manner. Therefore, the exchangeability assumption continues to hold given that they are originally exchangeable; CP is being applied exactly as intended. Yet the resulting sets are dramatically larger, revealing that score choice alone can create instability even under i.i.d. or exchangeable conditions. Intuitively, different scores exhibit different sensitivities as the effective sample size or semantic representation changes, which explains the substantial variability we observe.
Motivated by this, our work aims to design a paraphrase-robust nonconformity score and evaluate its efficiency using existing CP frameworks. Our experiments show that the resulting scores consistently outperform conventional ones, yielding smaller prediction sets, better calibration, and coverage rates that are more aligned with the target level.

## Valid coverage guarantee even under distributional shift
Our proposed score is inherently stochastic: it aggregates multiple independently generated paraphrases to produce a more robust nonconformity score. Because this paraphrasing-and-aggregation procedure is applied **identically and independently** to every input example, it **does not alter the exchangeability** of the original data. In fact, it can even restore exchangeability when the dataset is not exchangeable to begin with.
The key insight is that paraphrasing is **distributionally closed**: paraphrasing a question once or twice yields semantically equivalent variants that can be regarded as draws from the same underlying distribution. As a result, if there is a meaning-preserving semantic shift between the test data and the calibration data—where one has been reworded by an LLM and the other has not—our score still preserves score exchangeability. This is because our procedure always _paraphrases every input example_, regardless of whether the original data was reworded or not, and therefore forces all scores to follow the same paraphrased-once distribution. As a result, we re-establish score-level exchangeability even under shifts caused purely by paraphrasing. In the revised paper, we formalize this argument and provide a mathematical statement of distributional closedness in Section 3.3.

---

> ### Author Response · Authors · 2025-11-22
> **Overall Response (Part 2/2)**
>
> As a clarification, our initial draft used the term “_adversarial_’’ to refer to the case where both calibration and test data are paraphrased. This terminology understandably caused confusion, so we now refer to this setting as “fully reworded’’. We additionally introduce a new “semi-reworded’’ setting, where only the test input questions are reworded by an LLM, but the train/calibration set uses the human-written input questions, to further illustrate that our score maintains near-nominal coverage even under such semantic shifts. In contrast, baseline methods exhibit significant degradation in coverage and substantial inflation in prediction set size.
>
>
> ## The contributions and takeaways
> With the above clarification and the new theoretical support, we restate our contributions in a clearer way:
> **(1) A new empirical finding: paraphrase-sensitive CP failure.**
>  We introduce a simple “fully reworded” setup where both test data and calibration data are reworded by an LLM without changing their semantic meaning. Even though exchangeability still holds, standard CP scores can still lead to 2–4× larger prediction sets. This reveals a robustness issue that existing CP-for-LLM work has not examined.
> **(2) Paraphrase-aware, data-driven nonconformity scores.**
> We propose Paraphrase-Aware (PA) scores that aggregate multiple independently generated paraphrases and incorporate semantic correctness through a lightweight proxy model. These scores are designed to reduce surface-form variability in LLM outputs while preserving the distribution-free guarantees of standard CP/QCCP. Across a large-scale evaluation spanning seven MCQA datasets, three LLM families, and two independent LLM paraphrase generators, PA scores consistently achieve near-nominal coverage, smaller prediction sets, and more uniform class-wise coverage. Ablation studies further confirm that both components—multiple paraphrases and the proxy model—are necessary and complementary.
> **(3) A general theoretical framework for augmentation-based score design.**
> While our empirical focus is on paraphrasing, we introduce a more general theoretical framework (Sec. 3.3) showing that conformal validity can be preserved under any **distributionally closed** transformation. Proposition 3.1 establishes that applying such a transformation uniformly to calibration and test inputs restores score-level exchangeability even under meaning-preserving semantic shift. Lemma 3.1 further shows that our PA scores inherit this symmetry because they are deterministic, permutation-invariant functions of independently transformed inputs. Paraphrasing is a natural and practically relevant instance of this class, and this theory yields a principled augmentation-based approach: whenever a meaning-preserving or invariant transformation is available, applying it consistently ensures that optimized, data-driven scores remain exchangeable and thus maintain CP/QCCP coverage guarantees.
>
> Our work provides an example where the score function can be directly optimized by minimizing a suitable loss function, and thanks to the distribution-free nature of existing CP frameworks, the resulting score can be used with valid guarantees. Another insight is that once we know the form of meaning-preserving semantic shift, we can apply simple random post-processing or data augmentation (paraphrasing in our case) to mitigate the shift so that score exchangeability still holds.
>
> ## Summary of Revisions and Additional Experiments
>
> We thank the reviewers for their insightful suggestions. In response, we have made substantial
> revisions and added new experiments. All changes in the updated manuscript are highlighted in blue.
>
> **Major Updates of Paper Content**
> - Revised Figure 1 to more clearly illustrate paraphrase sensitivity and the improvement from PA scores.
> - Strengthened the theoretical section, with a formal analysis of distributional closedness and score exchangeability across normal, fully reworded, and semi-reworded settings.
> - Added rebuttal-specific experiments to the appendix, and a list of experiments is listed below.
> - Expanded related work in the appendix, including rationale for baseline selection and discussion of semantic-entropy and perturbation-based UQ methods.
> - Clarified future directions, including extensions to free-form generation and factuality tasks.
>
> **Rebuttal-Specific Experiments**
> - Semi-reworded setting results using another independent paraphrase generator and multiple datasets.
> - Full-reworded results using an alternative paraphraser to test cross-model robustness.
> - Dataset-level paraphrase-quality evaluation.
> - Runtime breakdown across all components.
> - Alternative embedding/pooling ablations.
> - Multi-seed confidence intervals for key figures.
>
> We hope the above explanation addresses the key concerns and helps reviewers better appreciate our work. For the remaining points, we provide point-by-point responses below.

---

### Author Response · Authors · 2025-11-30
**Summary of Reviewer Feedback During Discussion (Part 1/2)**

Dear Area Chair,

We sincerely appreciate your time and effort in handling our submission. To support your assessment, we provide a concise summary of the key developments during the rebuttal and discussion period. We hope this overview is helpful, and we would be glad to clarify anything further if needed.

---

### Summary of Discussion Outcome

**Three reviewers** (iUNn, 9pty, Kxds) indicated that **our responses adequately addressed their concerns**. **Two reviewers** (iUNn and 9pty) voluntarily **raised their scores** following the discussion. Overall, the reviewers expressed that the paper was substantially improved in clarity, motivation, and experimental rigor after the revisions and clarifications.


---

### Reviewer Engagement and Score Updates Before the Rollback

**Reviewer iUNn** confirmed that supplemental explanations fully resolved the earlier concerns. The reviewer raised the score from **4 → 8** and commented that the adversarial attack aspect opens up a meaningful future direction.


**Reviewer 9pty** acknowledged improvements to the manuscript and appreciated the clarifications on the distributional-closedness assumption. The reviewer increased the score from **2 → 4**, remarking that while the work is primarily experimental, its contributions are strengthened.


**Reviewer Kxds** maintained a positive score (**6**) and posted a follow-up comment thanking the authors for the thorough responses, stating that all questions were satisfactorily resolved.


**Reviewer vAQA**, whose score was 4, did not participate in the discussion phase.


---

Below, we provide a comprehensive summary of all reviewer questions and how each was addressed. If you would like to see the full, detailed responses, they are included in the sections that follow.

Thank you again for your time and consideration.

Sincerely,

Authors

---

> ### Author Response · Authors · 2025-11-30
> **Summary of Reviewer Feedback During Discussion (Part 2/2)**
>
> ### Summary of Reviewer Feedback and How We Addressed Them
>
> #### 1. Clarifying the Paper’s Core Contribution
>
> **Concerns raised by:** Kxds, vAQA, iUNn, 9pty
> **Reviewer questions:**
> - Unclear what is new vs. classical CP.
> - Method appears incremental or primarily empirical.
> - Need clearer articulation of contribution and motivation.
>
> **How we addressed it:**
> - Explicitly separated classical CP from our novel components (paraphrase-aware score design), and added comparison with related work in Appendix F.6.
> - Revised the Introduction section to include a clear contributions list:
>   1. New empirical phenomenon (paraphrase-induced CP set-size inflation).
>   2. New paraphrase-aware, data-driven nonconformity scores.
>   3. New theoretical framework (distributional closedness + exchangeability restoration).
> - Revised Figure 1 and the introduction to better motivate the problem.
>
> ---
>
> #### 2. Theoretical Validity and Exchangeability Under Paraphrasing
> **Concerns raised by:** Kxds, vAQA, 9pty, iUNn
> **Reviewer questions:**
> - Why does paraphrase aggregation preserve coverage?
> - Does paraphrasing break exchangeability, especially if only test data are reworded?
> - Is the distributional-closedness assumption too strong?
>
> **How we addressed it:**
> - Added Section 3.3 and Appendix A.2 with:
>   - Assumption 3.1 (distributional closedness)
>   - Proposition 3.1 (transformation restores exchangeability)
>   - Lemma 3.1 (PA scores preserve score-level exchangeability)
> - Clarified that paraphrasing preserves exchangeability, ensuring CP validity.
> - Added semi-reworded experiments in Appendix F.5, where calibration and test datasets have input-level rewording-induced shift, but PA score computation removes this mismatch, confirming theoretical predictions.
> - Explained that distributional closedness is realistic for meaning-preserving paraphrasing and not intended as a universal shift assumption.
>
> ---
>
> #### 3. Robustness to Paraphrase Generator, Embeddings, Error Bars, and Paraphrase Count
> **Concerns raised by:** Kxds, vAQA, iUNn
> **Reviewer questions:**
> - Risk of bias from using one paraphrase generator.
> - Are paraphrases semantically consistent?
> - Need to add error bars to the plots.
> - Sensitivity to embedding choice or number of paraphrases.
>
> **How we addressed it:**
> - Re-ran all paraphrase experiments using both Qwen and Llama, and results remain stable in Appendix F.4.
> - Added semantic similarity evaluation (SBERT, BLEU, ROUGE-L) for paraphrase quality in Appendix F.7.
> - Added error bars with 5 different random seeds to all the plots in the updated PDF.
> - Added embedding ablations (last hidden, mean, attention-weighted, intermediate-layer) in Appendix F.9.
> - Emphasized existing ablation in Table 5 showing improved robustness with more paraphrases, but small k already works.
>
> ---
>
> #### 4. Computational Efficiency and Overhead
> **Concerns raised by:** Kxds, vAQA
> **Reviewer questions:**
> - How much time does paraphrasing add?
> - Is inference-time overhead substantial?
>
> **How we addressed it:**
> - Added full wall-clock runtime breakdown in Appendix F.8.
> - Clarified that inference-time overhead is minimal and that fewer paraphrases still yield strong performance.
>
> ---
>
> #### 5. Methodological Clarifications (Data Leakage, Score Variants, Proxy Model Stability, Paraphrase Boundaries)
> **Concerns raised by:** Kxds, vAQA, iUNn
> **Reviewer questions:**
> - Do paraphrases introduce data leakage?
> - Why does the "worst" score perform poorly?
> - Why use a proxy model instead of calibrating LLM logits?
> - How to ensure the paraphrase generator produces diverse yet semantically-invariant paraphrases?
> - What defines the boundary of a valid paraphrase?
>
> **How we addressed it:**
> - Clarified that paraphrases are used only within the PA score computation and do not enter the calibration set, so there is no leakage.
> - Added discussion in Appendix F.10 to analyze the properties of three nonconformity score variants, clarifying the limitations of the worst-case score.
> - Explained that LLM projection heads are miscalibrated for MCQA; proxy model is lightweight and avoids modifying the LLM.
> - Explained that a set of paraphrasing prompts are used to request different wording while explicitly instructing the model to preserve meaning.
> - Added semantic similarity analysis in Appendix F.7 and clarified the practical scope of meaning-preserving paraphrasing.
>
> ---
>
> #### 6. Scope and Applicability Beyond MCQA
> **Concerns raised by:** vAQA, 9pty
> **Reviewer questions:**
> - Does the method extend beyond MCQA to free-form generation?
> - Is the work primarily experimental?
>
> **How we addressed it:**
> - Added discussion in Appendix F.11 on applying PA scores to factuality verification via claim decomposition.
> - Clarified that the empirical focus is MCQA, but the theoretical framework applies broadly to any transformation satisfying distributional closedness.

---

### Meta-Review · Area_Chair_jM73 · 2026-01-07

**Summary:**

The reviewers broadly agreed that the paper tackles an important and timely problem: the instability of conformal prediction–based uncertainty quantification for LLMs under paraphrase variation. Strengths consistently noted include the clear empirical demonstration of paraphrase sensitivity in existing CP scores, the intuitive design of paraphrase-aware (PA) nonconformity scores, and extensive experiments across datasets and model families. However, concerns centered on the novelty and depth of the contribution, the lack of theoretical justification for why paraphrase aggregation preserves CP guarantees, the reliance on LLM-generated paraphrases (with potential bias and cost), the limited baselines, and the missing runtime analysis.

**Reviewer Concerns:**

Concerns addressed by the rebuttal:
- Clarification of “adversarial” vs. reworded settings: Renaming and formalizing “fully reworded” and adding a “semi-reworded” setting resolves confusion raised by multiple reviewers about exchangeability and distribution shift.
- Contribution clarity: The authors separate the empirical discovery (paraphrase-sensitive CP failure) from the methodological contribution (paraphrase-aware score design).
- Runtime and efficiency: Added runtime breakdowns and cost analysis are presented.
- Paraphrase quality and bias: New paraphrase-quality evaluations and experiments using independent paraphrase generators are provided.
- Baselines and robustness: Expanded related work discussion and additional experiments partially address concerns about limited baselines and robustness validation.

Concerns still partially or fully outstanding:
- Scope of tasks: The method remains limited to MCQA; extensions to free-form generation are discussed but not demonstrated, leaving vAQA’s and others’ concerns only partially addressed.
- Perceived incremental nature: While better justified, some reviewers (notably 9pty) may still view the contribution as incremental relative to existing CP frameworks.
- Worst-case score behavior: Although discussed, the counterintuitive behavior of the “worst” aggregation variant may still be underexplained.
- Comparison to more recent UQ methods: While discussed in the appendix, the lack of direct empirical comparison may remain a concern.

**Reviewer Scores:**

Reviewer Kxds left comments without changing the original score. Reviewer iUNn and Reviewer 9pty mentioned that their original scores have increased, but the changed scores cannot be officially checked in the system.

---

### Decision · Program_Chairs · 2026-01-26

Reject